# A secreted microRNA disrupts autophagy in distinct tissues of *Caenorhabditis elegans* upon ageing

Yifei Zhou [1], Xueqing Wang[1], Mengjiao Song[1], Zhidong He[1], Guizhong Cui [1], Guangdun Peng [2,3], Christoph Dieterich[4], Adam Antebi[5,6], Naihe Jing [1] & Yidong Shen [1]*

Macroautophagy, a key player in protein quality control, is proposed to be systematically impaired in distinct tissues and causes coordinated disruption of protein homeostasis and ageing throughout the body. Although tissue-specific changes in autophagy and ageing have been extensively explored, the mechanism underlying the inter-tissue regulation of autophagy with ageing is poorly understood. Here, we show that a secreted microRNA, *mir-83/* miR-29, controls the age-related decrease in macroautophagy across tissues in *Caenorhabditis elegans*. Upregulated in the intestine by *hsf-1/*HSF1 with age, *mir-83* is transported across tissues potentially via extracellular vesicles and disrupts macroautophagy by suppressing CUP-5/MCOLN, a vital autophagy regulator, autonomously in the intestine as well as non-autonomously in body wall muscle. Mutating *mir-83* thereby enhances macroautophagy in different tissues, promoting protein homeostasis and longevity. These findings thus identify a microRNA-based mechanism to coordinate the decreasing macroautophagy in various tissues with age.

[1] State Key Laboratory of Cell Biology, Innovation Center for Cell Signaling Network, CAS Center for Excellence in Molecular Cell Science, Shanghai Institute of Biochemistry and Cell Biology, University of Chinese Academy of Sciences, Chinese Academy of Sciences, 320 Yueyang Rd, 200031 Shanghai, China. [2] CAS Key Laboratory of Regenerative Biology, Guangdong Provincial Key Laboratory of Stem Cell and Regenerative Medicine, Guangzhou Institutes of Biomedicine and Health, Chinese Academy of Sciences, 510530 Guangzhou, China. [3] Guangzhou Regenerative Medicine and Health Guangdong Laboratory (GRMH-GDL), 510005 Guangzhou, China. [4] Klaus Tschira Institute for Integrative Computational Cardiology and Department of Internal Medicine III, University Hospital Heidelberg, Neuenheimer Feld 669, 69120 Heidelberg, Germany. [5] Max Planck Institute for Biology of Ageing, Joseph-Stelzmann-Str. 9b, D-50931 Cologne, Germany. [6] Cologne Excellence Cluster on Cellular Stress Responses in Aging-Associated Diseases (CECAD), University of Cologne, 50674 Cologne, Germany. *email: yidong.shen@sibcb.ac.cn

Ageing results from the disruption of homeostasis, such as protein homeostasis (proteostasis). Proteostasis is the functional balance of the proteome of a living organism. Its disruption is closely associated with age and leads to various ageing phenotypes and ageing-related disorders[1]. One of the critical factors in maintaining proteostasis is macroautophagy (hereafter referred to as autophagy)[1]. During autophagy, cytosolic cargo (e.g., protein aggregates) is engulfed into autophagosomes and degraded in autolysosomes after autophagosomes fuse with lysosomes[2]. Autophagy is enhanced in various longevity models, whereas multiple lines of evidence suggest a gradual dysregulation of autophagy with ageing, impairing proteostasis in aged animals[3,4]. However, the mechanism underlying the decrease in autophagy has scarcely been studied.

An intriguing feature of the decreasing proteostasis with ageing is that it is systematically coordinated in tissues with distinct proteomes and undergoing disparate stresses. Consistently, two crucial prsocesses for proteostasis (i.e., the unfolded protein response and the chaperone signaling response) function in a tissue non-autonomous manner when challenged with tissue-specific perturbations, such as polyglutamine in neurons and misfolded myosin heavy chain in muscle[5–9]. The ageing-induced impairment of autophagy is also systematically controlled among tissues[10], implying that tissues engage in crosstalk with each other to coordinate their autophagic activities. Previous studies on autophagy in tissues have focused on tissue-specific changes in autophagy and their functions in ageing[3]. However, little is known about the inter-tissue regulation of autophagy.

MicroRNAs are important regulators in various biological processes, including ageing and autophagy[11,12]. They are not only present in cells but have also been found to be secreted into different body fluids. Secreted microRNAs are widely proposed to be biomarkers of various patho-physiological conditions, including ageing[13,14]. Herein, we report a secreted microRNA, mir-83/miR-29, as a cross-tissue regulator of the decrease in autophagy with age. Upregulation of mir-83 in aged intestine by hsf-1/HSF1 impairs intestinal autophagy through inhibition of the mir-83 target CUP-5/MCOLN. Moreover, this autophagy regulation mechanism in the intestine also controls autophagy in body wall muscle (BWM). mir-83 is transported from the intestine into BWM potentially via extracellular vesicles, suppressing autophagy there by targeting BWM cup-5 per se. Our results thus reveal a secreted microRNA-based mechanism dysregulating autophagy in distinct tissues with ageing.

## Results

**mir-83 is induced by hsf-1 during ageing in the intestine**. To explore the mechanism impairing autophagy among tissues with ageing, we examined transcriptome changes in *Caenorhabditis elegans* at different ages. Wild-type (WT) worms at four stages of adulthood were analyzed: day 1 (D1, the start of adulthood), day 7 (D7, around the end of the reproductive period), day 14 (D14), and day 21 (D21, around the mean lifespan of WT worms). From microRNA-Seq analysis and qRT-PCR validation, *mir-83*, homologous to mammalian miR-29, was found to increase with ageing[15] (Fig. 1a and Supplementary Fig. 1a-b). A GFP reporter driven by the native promoter of *mir-83* confirmed the upregulation of this microRNA with ageing (Fig. 1b). Based on previous reports and our observations, *mir-83* is mostly expressed in the intestine and neurons (Fig. 1c and Supplementary Fig. 1c-d)[16,17]. Intriguingly, we observed significant induction of *mir-83* only in the intestine during ageing but not in neurons (Fig. 1c, d and Supplementary Fig. 1c-d), indicating that *mir-83* is upregulated specifically in the intestine by ageing.

We next questioned which transcription factor controls *mir-83* expression during ageing. *mir-83* has been reported to be upregulated by heat shock[18] and harbors heat shock elements in its promoter (Supplementary Fig. 1e)[19]. *hsf-1*/HSF1 is the master transcription factor of the heat shock response[20]. Upon using genome editing to tag endogenous *hsf-1* with GFP::3xFLAG, we observed increases in HSF-1 protein levels in worms at D4 and D10 (Supplementary Fig. 1f). Consistent with the specific upregulation of *mir-83* in the aged intestine, further analysis of the fluorescent signal from HSF-1:: GFP::3xFLAG in live worms indicated that HSF-1 was upregulated in the intestines but not in the neurons of aged animals (Supplementary Fig. 1g). We then examined whether *hsf-1* controls *mir-83* during ageing. Suppressing *hsf-1* by RNAi completely blocked the upregulation of *mir-83* expression in worms at D4 and D7 but had little effect on *mir-83* levels at D1 (Fig. 1e and Supplementary Fig. 1h). Accordingly, *hsf-1* RNAi suppressed the GFP reporter of *mir-83* transcription in the intestine at D7 but not at D1 (Fig. 1f). Taken together, these data indicate that *hsf-1* promotes the expression of *mir-83* in the aged intestine.

**cup-5/MCOLN is a target of mir-83**. MicroRNAs function through binding to the 3′-UTRs of target genes mRNA and inhibiting their translation. By TargetScan[21], two autophagy-related genes, *cup-5*/MCOLN and *atg-4.2*/ATG4D[22,23], were predicted as potential targets of *mir-83* (Supplementary Fig. 2a). Transfection of *mir-83* mimic, but not a control oligo, into HEK293T cells inhibited the expression of the luciferase reporter with the 3′-UTR of either *cup-5* or *atg-4.2* (Fig. 2a and Supplementary Fig. 2b). When the *mir-83* target site in the *cup-5* or *atg-4.2* 3′-UTR was mutated, the *mir-83* mimic no longer inhibited the expression of the luciferase reporter of the 3′-UTR (Fig. 2a and Supplementary Fig. 2b). These results therefore demonstrate the interactions of *mir-83* with these two genes. To confirm their interactions *in vivo*, we followed our previous reports and prepared transgenic animals with dual fluorescence, expressing mCherry with the native promoter and the 3′-UTR of *cup-5* or *atg-4.2*, and a ubiquitously expressed GFP as an internal control[24,25]. If *mir-83* interacts with the 3′-UTR of interest *in vivo*, the ratio of mCherry versus GFP (mCherry/GFP) would be expected to increase in *mir-83(-)* mutants (with deletion of the *mir-83* gene) (Fig. 2b). As expected, the dual fluorescent reporter of *cup-5* showed an elevated mCherry/GFP ratio upon *mir-83* deletion (Fig. 2c). However, the mCherry/GFP ratio of the *atg-4.2* 3′-UTR reporter was unchanged in *mir-83(-)* mutants (Supplementary Fig. 2c-d), indicating that *mir-83* does not interact with the 3′-UTR of *atg-4.2 in vivo*.

We next assessed the protein level of CUP-5. Upon using an extrachromosomal transgene of mCherry::*cup-5* controlled by its native promoter and 3′-UTR, we observed an elevated level of mCherry::CUP-5 in the intestines of *mir-83(-)* mutants at different ages (Fig. 2d). We further knocked in a 3xFLAG:: WrmScarlet tag at the 5′ of the *cup-5* gene with CRISPR/Cas9 technology for more precise quantification of endogenous CUP-5 levels. Consistent with the upregulation of mCherry:: CUP-5 in *mir-83(-)* mutants (Fig. 2d and Supplementary Fig. 2f), the endogenously labeled CUP-5 was also increased in *mir-83(-)* mutants at different ages (Fig. 2e). Therefore, *mir-83* inhibits CUP-5. The mRNA level of *cup-5* was unchanged with age, with a mild but non-significant increase upon *mir-83* mutation (Supplementary Fig. 2e), implying that *mir-83* inhibits *cup-5* mainly by blocking its translation but not by degrading its mRNA. Collectively, the results show that *cup-5* is a target of *mir-83*.

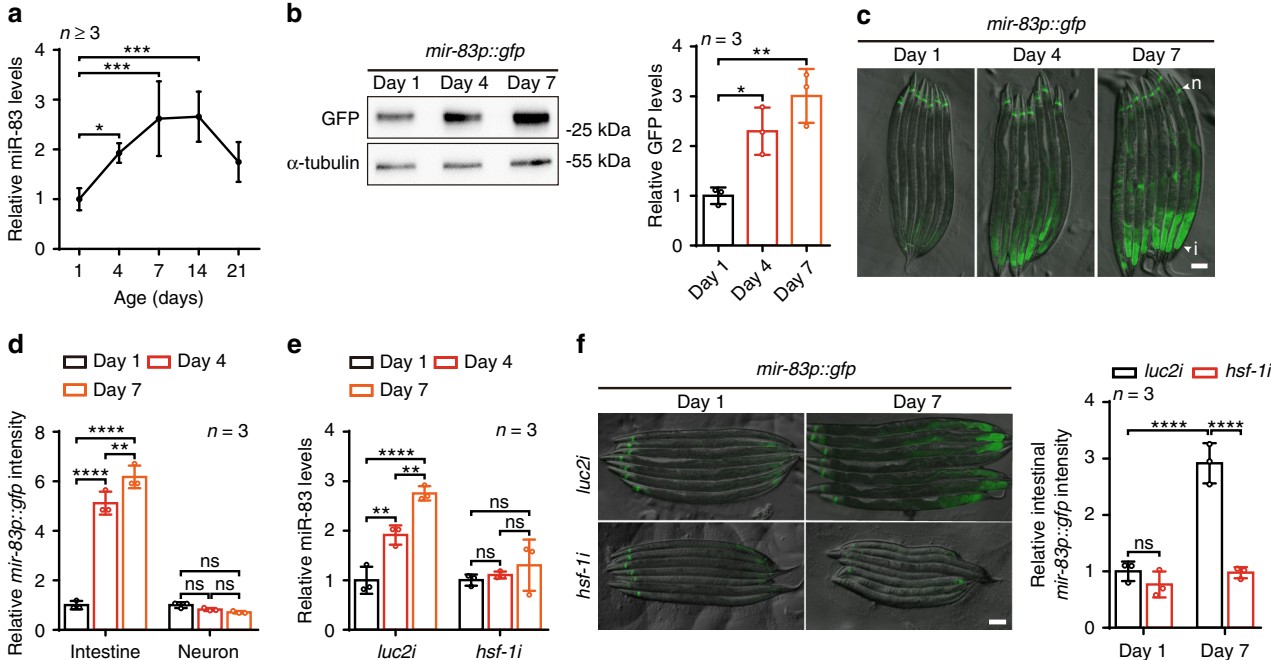

**Fig. 1** Ageing upregulates *mir-83* in the intestine through *hsf-1*. **a** qRT-PCR of miR-83 in the wild type (WT) worms of indicated ages. The levels of miR-83 were normalized against the one at day 1. *n* = 7, 4, 3, 3, and 3 independent experiments for samples of day 1, 4, 7, 14, and 21, respectively. **b** Immunoblots of *mir-83p*::GFP at indicated ages. Blots against α-tubulin and day 1 sample serve as controls for loading and normalization respectively. *n* = 3 independent experiments. **c** Representative images of *mir-83p::gfp* transgenic animals at indicated ages. n: neuron, i: intestine. **d** GFP intensity of *mir-83p::gfp* in the intestine and head neurons at day 1, day 4, and day 7 of adulthood. Samples of day 1 serve as controls for normalization. *n* = 3 independent experiments containing at least 41 worms. **e** qRT-PCR of miR-83 in WT worms treated with *luc2* or *hsf-1* RNAi. RNAi treatment was performed from hatching and worms were examined at indicated ages. Samples of day 1 serve as controls for normalization. *n* = 3 independent experiments. **f** Expression of *mir-83p::gfp* under *luc2* or *hsf-1* RNAi. Worms were treated with RNAi from hatching and examined for GFP intensity at indicated ages. day 1 worms subjected to *luc2* RNAi serve as control for normalization. *n* = 3 independent experiments containing at least 39 worms. Scale bars: 100 μm. Statistical significance was calculated by One-way ANOVA in (**a** and **b**), or Two-way ANOVA in **d**–**f**. ns: non-significant, *$p < 0.05$, **$p < 0.01$, ***$p < 0.001$, ****$p < 0.0001$. Source data are provided as a Source Data file

***mir-83* impairs intestinal autophagy through *cup-5*.** The target of *mir-83*, *cup-5*/MCOLN, encodes a calcium channel localized to the lysosomal membrane and controls multiple steps of autophagy[22,26,27]. Therefore, *mir-83* could control autophagy by inhibiting CUP-5. As *mir-83* is upregulated in the aged intestine (Fig. 1c-d and Supplementary Fig. S1c-d), we first analyzed autophagy in this tissue. GFP::LGG-1 (GFP-LC3)-labeled autophagosomes (APs) are well-recognized autophagy markers[28]. Indeed, more GFP::LGG-1 puncta (APs) were observed in the intestines of *mir-83(-)* mutants than in the intestines of WT worms (Fig. 3a). Increases in APs could be due to either induction or inhibition of autophagy[28]. We then treated worms with chloroquine to distinguish between these two possibilities. When autophagy is active, chloroquine treatment increases AP numbers because it inhibits the transition of APs to autolysosomes (ALs) and the degradation of protein in ALs associated with active autophagy flux[29]. When autophagy is already blocked, chloroquine treatment can no longer change the number of APs. At 1 h post treatment with chloroquine, the numbers of GFP::LGG-1-labeled APs were increased in the intestines of both the WT worms and the *mir-83(-)* mutants (Fig. 3b), indicating that autophagy flux was not blocked upon mutation of *mir-83*. Thus, mutation of *mir-83* upregulates AP numbers to promote autophagy.

We next used mCherry-GFP-tagged LGG-1 to further examine the effect of *mir-83* on autophagy. GFP, but not mCherry, in this reporter is quenched in ALs, making APs and ALs respectively visible as yellow and red puncta[10]. In the intestines of *mir-83(-)* mutants, the numbers of both APs and ALs were increased

(Fig. 3c), in accordance with our previous observation (Fig. 3a, b). We also validated these results with intestine-specific dual GFP (dFP)-tagged LGG-1, which releases monomeric GFP (mFP) in ALs. The mFP/dFP ratio correlates with the activity of autophagy[30]. Consistent with our observations with other autophagy metrics, increased mFP/dFP ratios were detected in *mir-83(-)* mutants (Supplementary Fig. 3a).

We next examined autophagy in neurons, where *mir-83* is enriched but not upregulated by ageing (Fig. 1c, d and Supplementary Fig. 1c-d). A decrease in SQST-1::GFP (P62-GFP) puncta, which are substrates for autophagy, represents an increase in autophagy[28]. The transcription of *sqst-1* is not under the regulation of ageing or *mir-83* (Supplementary Fig. 3b). In the head neurons of *mir-83(-)* mutants, the SQST-1::GFP puncta were decreased modestly by 27% (Fig. 3e). However, when examined with a neuron-specific dFP-LGG-1 reporter, the mFP/dFP ratio was unchanged upon mutation of *mir-83* while controlled by chloroquine (Fig. 3f). Further analysis using a neuron-specific mCherry::GFP::LGG-1 reporter indicated that neuronal autophagy responded to chloroquine treatment but was not significantly changed in *mir-83(-)* mutants (Fig. 3g, h). Therefore, *mir-83* could have weak control over autophagy in neurons.

To determine whether *mir-83* controls autophagy through *cup-5*, *cup-5*-related autophagy phenotypes in WT worms and *mir-83 (-)* mutants were first examined. If *mir-83* dysregulates autophagy by inhibiting CUP-5, RNAi against *cup-5* should suppress the increase in CUP-5 and abolish the enhancement of *cup-5*-related autophagy phenotypes in *mir-83(-)* mutants. The numbers of lysosomes were first determined, since *cup-5(-)* mutants exhibit

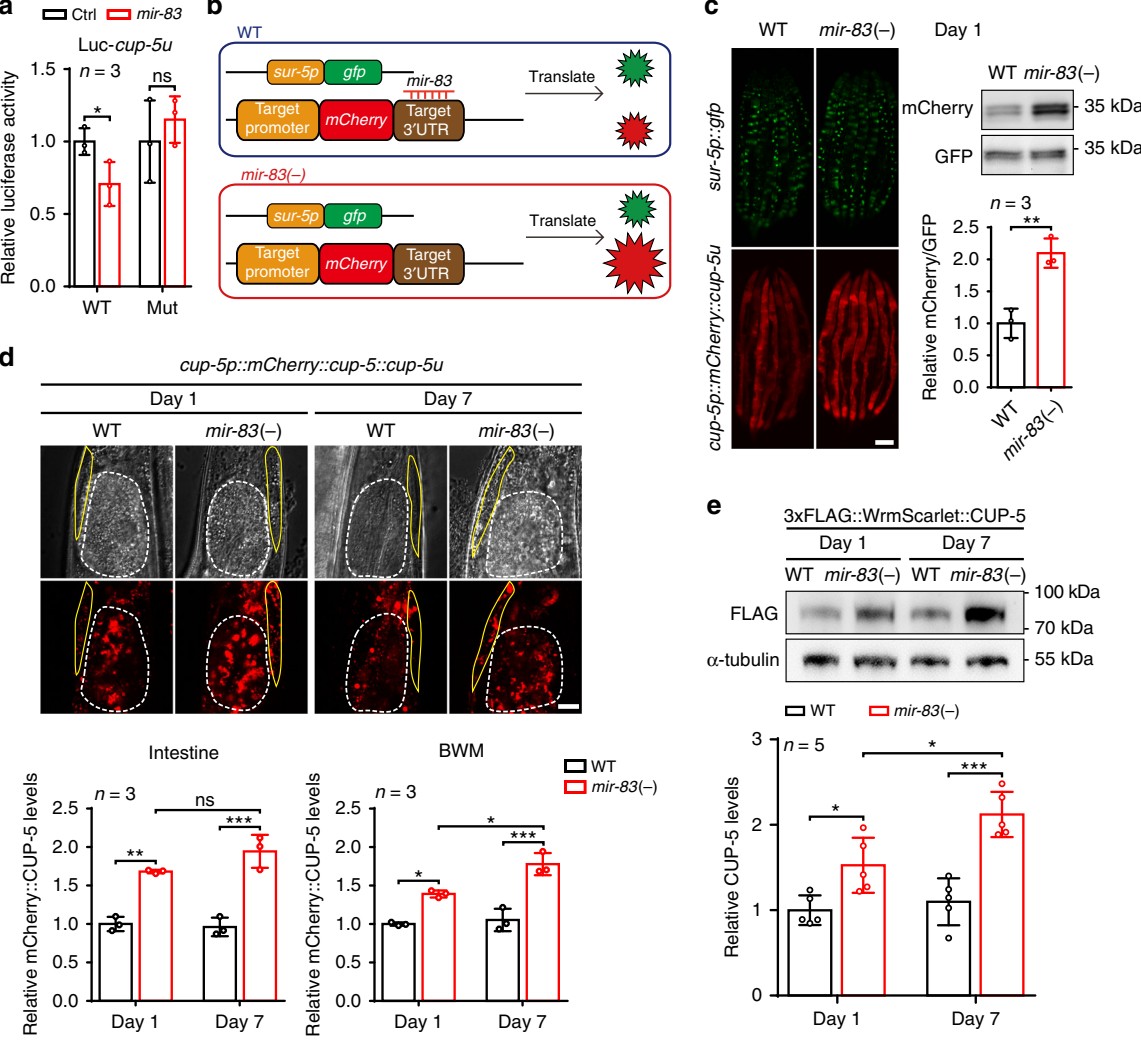

**Fig. 2** *mir-83* inhibits *cup-5* by binding to its 3′-UTR. **a** cel-mir-83-3p mimic (*mir-83*) but not a control oligo (Ctrl) suppresses a luciferase reporter with *cup-5* 3′-UTR (WT) in HEK293T cells. Ctrl transfected cells serve as controls for normalization. Mutating the *mir-83* binding site in *cup-5* 3′-UTR (Mut) blocked this interaction. *n* = 3 independent experiments. **b** A depiction of the dual fluorescence reporter to test the interaction of *mir-83* and its targets in vivo. **c** Fluorescent signals and immunoblots of the dual fluorescence reporter of *cup-5* 3′-UTR in WT worms and *mir-83(-)* mutants at day 1 of adulthood. Quantification is from the western blots. GFP blots and WT worms serve as controls for loading and normalization respectively. Scale bar: 100 μm. *n* = 3 independent experiments. **d** Fluorescent signals of the *cup-5* translational reporter (mCherry::CUP-5) in the posterior intestine (dashed lines) and nearby BWM (yellow lines) of WT worms and *mir-83(-)* mutants at indicated ages. WT signals at day 1 serve as controls for normalization. Scale bar: 10 μm. n = 3 independent experiments. **e** The protein level of the endogenously tagged 3xFLAG::WrmScarlet::CUP-5 is increased upon mutating *mir-83* at indicated ages. α-tubulin and WT blots at day 1 serve as controls for loading and normalization respectively. *n* = 5 independent experiments. Statistical significance was calculated by unpaired *t*-test in **a**, **c**, **e**, or two-way ANOVA in **d**. ns: non-significant, *$p < 0.05$, **$p < 0.01$, ***$p < 0.001$. Source data are provided as a Source Data file

decreased numbers of lysosomes due to impaired budding and maturation of lysosomes[22,31]. Consistently, we observed increases in the numbers of lysosomes in the intestines of *mir-83(-)* mutants, which was suppressed upon *cup-5* RNAi (Supplementary Fig. 3c-d). Therefore, *mir-83* controls lysosome biogenesis by targeting *cup-5*. We next examined the protein level of LGG-1/LC3, since the mammalian orthologue of *cup-5*, MCOLN, activates the transcription factor EB (TFEB) to promote the expression of LC3 and other key autophagy genes[26,27]. GFP::LGG-1 was indeed increased in *mir-83(-)* mutants (Supplementary Fig. 3e). RNAi against *cup-5* in *mir-83(-)* mutants restored the expression of GFP::LGG-1 to WT levels (Supplementary Fig. 3f), indicating that *mir-83* controls LGG-1 expression via *cup-5* and implying that the MCOLN-TFEB axis may be

conserved in *C. elegans*. Because LGG-1 and lysosomes are key to the formation of APs and ALs, we finally examined the numbers of APs and ALs in the intestines of WT worms and *mir-83(-)* mutants upon *cup-5* RNAi. Using the mCherry-GFP::LGG-1 reporter, we found that RNAi against *cup-5* completely inhibited the increases in ALs in the intestines of *mir-83(-)* mutants (Supplementary Fig. 3g), indicating that *cup-5* is required for the enhanced autophagy flux in *mir-83(-)* mutants. Intestine-specific RNAi treatment against *cup-5* abolished the elevated autophagy flux in the intestines of *mir-83(-)* mutants as well (Fig. 3d), indicating cell-autonomous regulation of *cup-5*-mediated autophagy in the intestine by *mir-83*. Taken together, *mir-83* regulates intestinal autophagy with ageing by downregulating *cup-5* cell-autonomously.

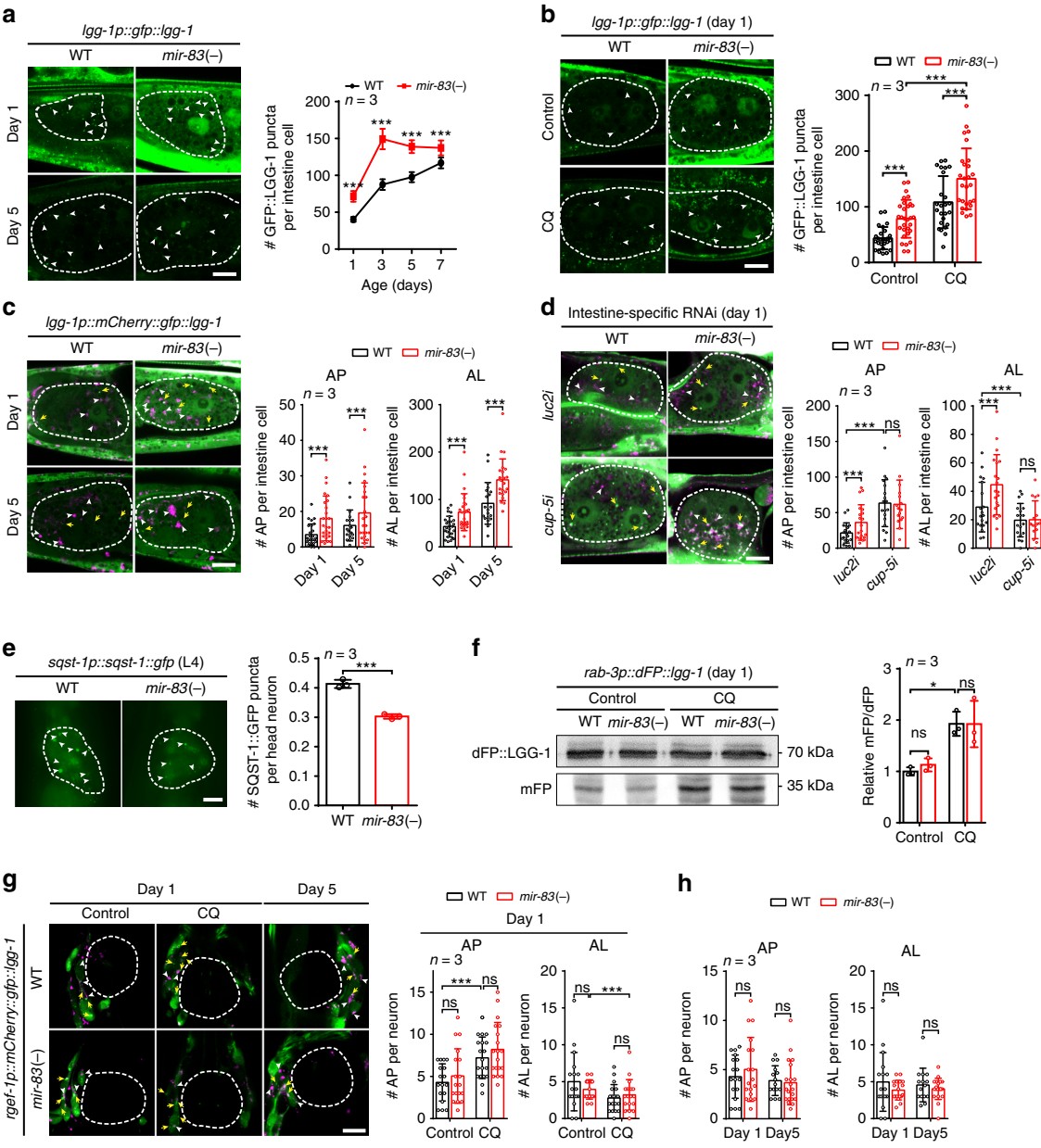

**Fig. 3** *mir-83* dysregulates autophagy in the intestine through *cup-5*. **a**, **b** GFP::LGG-1 puncta (arrow heads) in the posterior intestine cells (dashed lines) of the WT worms and *mir-83(-)* mutants at indicated ages. Compared to that of day 1 samples, the exposure of day 5 samples was increased to visualize GFP:: LGG-1 puncta. Worms were treated with 5 mM chloroquine (CQ) or a mock control **b**. $n = 3$ independent experiments containing at least 22 worms in **a** and 25 worms in **b**. **c** The fluorescent signals of mCherry-GFP::LGG-1 in the posterior intestine cells (dashed lines) of WT worms and *mir-83(-)* mutants at indicated ages. White arrow heads denote the autolysosome (AL) puncta with only mCherry signals. Yellow arrows denote the autophagosome (AP) puncta with both GFP and mCherry signals. $n = 3$ independent experiments containing at least 19 worms. **d** APs (yellow arrows) and ALs (white arrow heads) in the posterior intestine cells (dashed lines) of WT worms and *mir-83(-)* mutants. Animals were treated with intestine specific RNAi against *luc2* or *cup-5* from hatching and examined at day 1 of adulthood. $n = 3$ independent experiments containing at least 16 worms. **e** The number of SQST-1::GFP puncta (arrow heads) is modestly reduced in the head neurons (dashed lines) of *mir-83(-)* mutants at the fourth larval stage (L4). $n = 3$ independent experiments containing at least 58 worms. **f** The dFP::LGG-1 reporter (mFP/dFP, mFP vs dFP::LGG-1) in neurons responds to the treatment of 5 mM CQ but not *mir-83* mutation at day 1 of adulthood. WT samples with control treatment serve as controls for normalization. $n = 3$ independent experiments. **g–h** APs (yellow arrows) and ALs (white arrow heads) in the head neurons surrounding the posterior pharynx bulb (dashed lines). Worms of indicated genotypes were examined at day 1 and day 5 of adulthood. A CQ treatment (5 mM) was performed as indicated. $n = 3$ independent experiments containing at least 15 worms. Scale bars: 10 μm. Statistical significance was calculated by Poisson regression in **a–d** and **g–h**, unpaired *t*-test in **e**, or one-way ANOVA in **f**. ns: non-significant, $*p < 0.05$, $***p < 0.001$. Source data are provided as a Source Data file

**Intestinal *mir-83* disrupts autophagy in body wall muscle.** Although *mir-83* is not expressed in body wall muscle (BWM) (Fig. 1c and Supplementary Fig. 1c-d)[16,17], we unexpectedly found that autophagy in BWM is also under the control of *mir-83*. Upon examining the signals from GFP::LGG-1 and mCherry-

GFP::LGG-1, we observed increases in both APs and ALs in the BWM cells of *mir-83(-)* mutants (Fig. 4a, b). Chloroquine treatment increased the numbers of APs in BWM of WT worms and *mir-83(-)* mutants at D1 (Fig. 4c), indicating that autophagy is active rather than blocked in BWM upon *mir-83* mutation. We

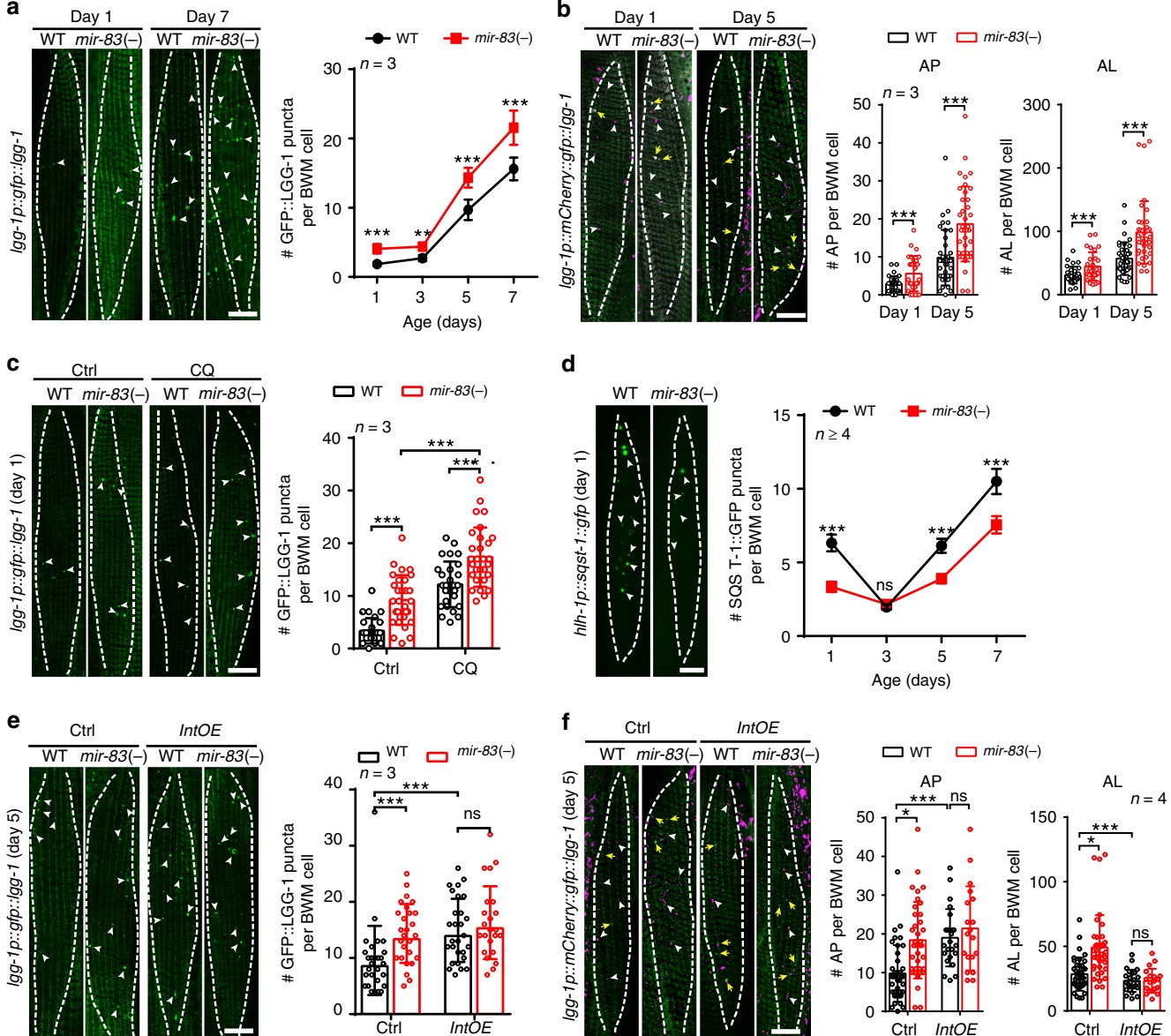

**Fig. 4** *mir-83* impairs autophagy in the body wall muscle with ageing. **a** GFP::LGG-1 puncta (arrow heads) in the body wall muscle (BWM) (dashed lines) of WT worms and *mir-83(-)* mutants at indicated ages. *n* = 3 independent experiments containing at least 25 worms. **b** mCherry::GFP::LGG-1 signals in the BWM (dashed lines) of WT worms and *mir-83(-)* mutants at indicated ages. White arrow heads denote the AL puncta and yellow arrows denote the AP puncta. *n* = 3 independent experiments containing at least 29 worms. **c** GFP::LGG-1 puncta (arrow heads) in the BWM (dashed lines) of indicated strains at day 1 of adulthood post the treatment of 5 mM chloroquine (CQ) or a mock treatment (Ctrl). *n* = 3 independent experiments containing at least 28 worms. **d** SQST-1::GFP puncta (arrow heads) in the BWM (dashed lines) of indicated strains. *n* = 5, 5, 4, and 4 independent experiments for samples of day 1, 3, 5, and 7, respectively. Each sample contains at least 106 worms. **e** The number of GFP::LGG-1 puncta (arrow heads) in WT worms and *mir-83(-)* mutants with or without an intestine-specific transgene of *mir-83* (*IntOE*). Dashed lines denote BWM cells. *n* = 3 independent experiments containing at least 23 worms. **f** The intestine-specific expression of *mir-83* abolishes the increased ALs (white arrow heads) in BWM cells (dashed lines) of *mir-83(-)* mutants. Note that APs (yellow arrows) were increased in WT worms due to a block of the autophagy flux. *n* = 4 independent experiments containing at least 19 worms. Scale bars: 10 μm. Statistical significance was calculated by Poisson regression. ns: non-significant, \*p < 0.05, \*\*p < 0.01, \*\*\*p < 0.001. Source data are provided as a Source Data file

further used a BWM-specific SQST-1::GFP transgene to confirm the elevated autophagy flux in *mir-83(-)* BWM[28]. As expected, SQST-1::GFP puncta in BWM were decreased in *mir-83(-)* mutants compared with WT worms (Fig. 4d).

Since *mir-83* is induced in the intestine with ageing and is not expressed in BWM (Fig. 1c and Supplementary Fig. 1c-d)[16,17], we hypothesized that intestinal *mir-83* may regulate autophagy in BWM. An intestine-specific *mir-83* transgene indeed restored the increased ALs to WT levels in the BWM of *mir-83(-)* mutants

while further increasing the APs in the WT worms to the levels in *mir-83(-)* mutants (Fig. 4e, f), indicating that intestinal *mir-83* suppresses the AP-AL transition in BWM. Therefore, the induced *mir-83* in the intestine is responsible for the disrupted autophagy flux in BWM during ageing.

*mir-83* regulates autophagy in the intestine via *cup-5* (Fig. 3), and *cup-5* is also expressed in BWM (Fig. 2d)[32]. We therefore speculated that the enhanced BWM autophagy in *mir-83(-)* mutants could occur through *cup-5* in BWM. Indeed, RNAi

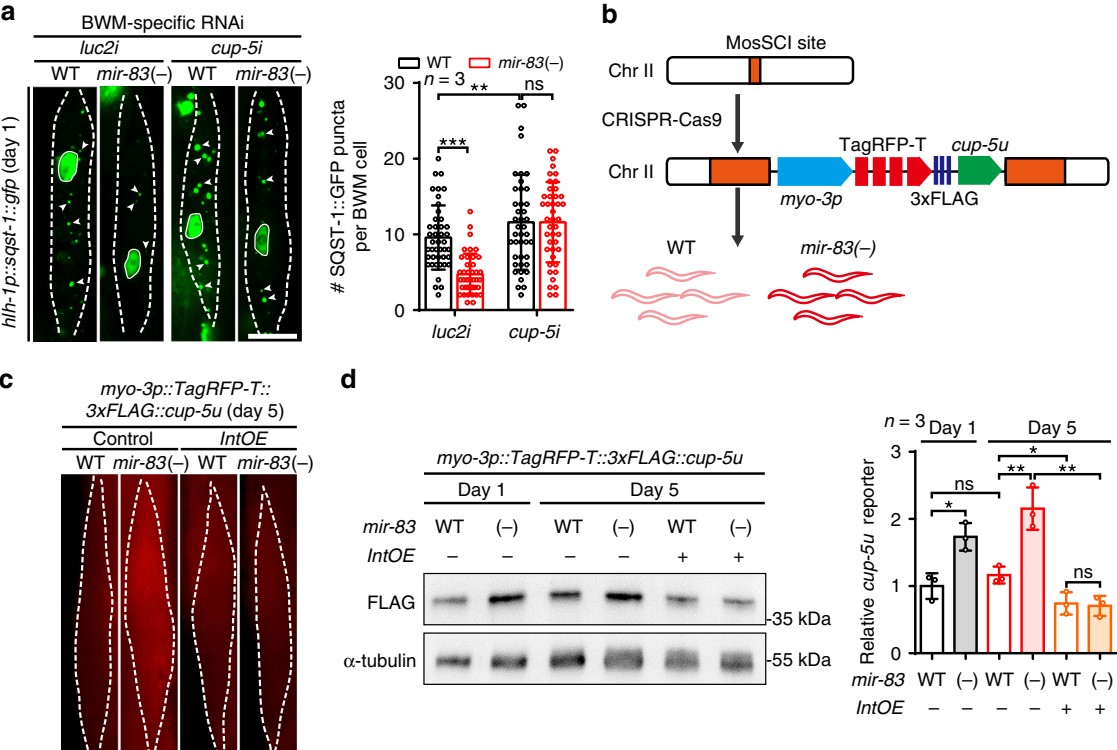

**Fig. 5** Intestinal *mir-83* inhibits *cup-5* in the body wall muscle. **a** SQST-1::GFP puncta (arrow heads) in the BWM cells (dashed lines) of indicated strains treated with BWM specific RNAi against *luc2* or *cup-5* from hatching. The GFP signals in nucleus were from the transgene enabling BWM specific RNAi. Worms were examined at day 1 of adulthood. *n* = 3 independent experiments containing at least 41 worms. **b** A diagram depicting an integrated transgene of a BWM-specific reporter of *cup-5* 3′-UTR. **c** Representative images of the BWM-specific *cup-5* 3′-UTR reporter in the indicated strains at day 5 of adulthood. *IntOE*: the intestine-specific overexpression of *mir-83*. Dashed lines denote myocytes. **d** The intestine-specific overexpression of *mir-83* (*IntOE*) abolishes the upregulation of the BWM-specific *cup-5* 3′-UTR reporter in *mir-83(-)* mutants. Blots against α-tubulin and WT worms at day 1 serve as controls for loading and normalization respectively. *n* = 3 independent experiments. Scale bars: 10 μm. Statistical significance was calculated by Poisson regression in **a**, or unpaired *t*-test in **d**. ns: non-significant, *\*p* < 0.05, *\*\*p* < 0.01, *\*\*\*p* < 0.001. Source data are provided as a Source Data file

against *cup-5* in various tissues or specifically in BWM restored the numbers of SQST-1::GFP puncta in the BWM of *mir-83(-)* mutants to WT levels (Fig. 5a and Supplementary Fig. 4a), indicating that *mir-83* regulates BWM autophagy in a *cup-5*-dependent manner.

To further test whether *cup-5* in BWM is under the direct regulation of *mir-83*, we inserted a BWM-specific *cup-5* 3′-UTR reporter into the genome using CRISPR/Cas9. If *mir-83* targets the 3′-UTR of *cup-5* in BWM, the reporter should be upregulated in *mir-83(-)* mutants (Fig. 5b). Indeed, mutation of *mir-83* increased this BWM-specific reporter by 50% at day 1 (D1) and further upregulated it by ~2-fold at day 5 (Fig. 5c, d). An extrachromosomal array expressing mCherry fused with the *cup-5* 3′-UTR and cyan fluorescent protein (CFP) specifically in BWM also showed an increase in the mCherry/CFP ratio upon mutation of *mir-83*, confirming the interaction between *mir-83* and the *cup-5* 3′-UTR in BWM (Supplementary Fig. 4b). The expression of mCherry::CUP-5 was also upregulated in the BWM of *mir-83(-)* mutants (Fig. 2d). Moreover, intestine-specific overexpression of *mir-83* suppressed the elevations in *cup-5* 3′-UTR reporter expression in the BWM of WT worms and *mir-83(-)* mutants (Fig. 5c, d), further indicating that *mir-83* from the intestine can suppress *cup-5* in BWM.

**mir-83 is transported across tissues.** Since *mir-83* directly interacts with the *cup-5* 3′-UTR in BWM but is not expressed in this tissue (Figs. 1c and 5 and Supplementary Figs. 1c-d and 4)[16,17], we speculated that *mir-83* could be transported into BWM from

other tissues, such as the intestine. To verify this hypothesis, we isolated BWM cells and performed single-cell qPCR (Fig. 6a). *rgef-1*, *vha-6*, and *myo-3* (three genes specifically expressed in neurons, intestine and BWM, respectively) were also examined in isolated BWM cells (Supplementary Fig. 5a-d). *rgef-1* and *vha-6* were not detected, therefore excluding contamination from the two *mir-83*-enriched tissues (i.e., the neurons and intestine). Indeed, *mir-83* was detected in BWM cells from WT animals but not in those from *mir-83(-)* mutants (Fig. 6b and Supplementary Fig. 5e-f). In the isolated BWM cells, *gfp* expressed from a BWM-specific transgene was detected, but TagRFP-T expressed by the *mir-83* promoter was not (Supplementary Fig. 5c-d), confirming that *mir-83* was not transcribed in BWM. In addition, BWM cells from *mir-83(-)* mutants carrying a transgene specifically overexpressing *mir-83* in the intestine also had elevated *mir-83* levels (Fig. 6b), whereas TagRFP-T driven by the same intestine-specific promoter was not detected in the isolated BWM cells (Supplementary Fig. 5c-d). Collectively, these data indicate that *mir-83* is transported from the intestine into BWM.

How is *mir-83* transported from the intestine into BWM? The pseudocoelom in *C. elegans* could mediate this transportation because it provides a medium for communication across tissues similar to the circulatory system in higher organisms and is located between the intestine and BWM. Coelomocytes in the pseudocoelom continually take in pseudocoelom fluids and are likely to be enriched with secreted *mir-83*[33]. Indeed, although *mir-83* is not expressed in coelomocytes (Supplementary

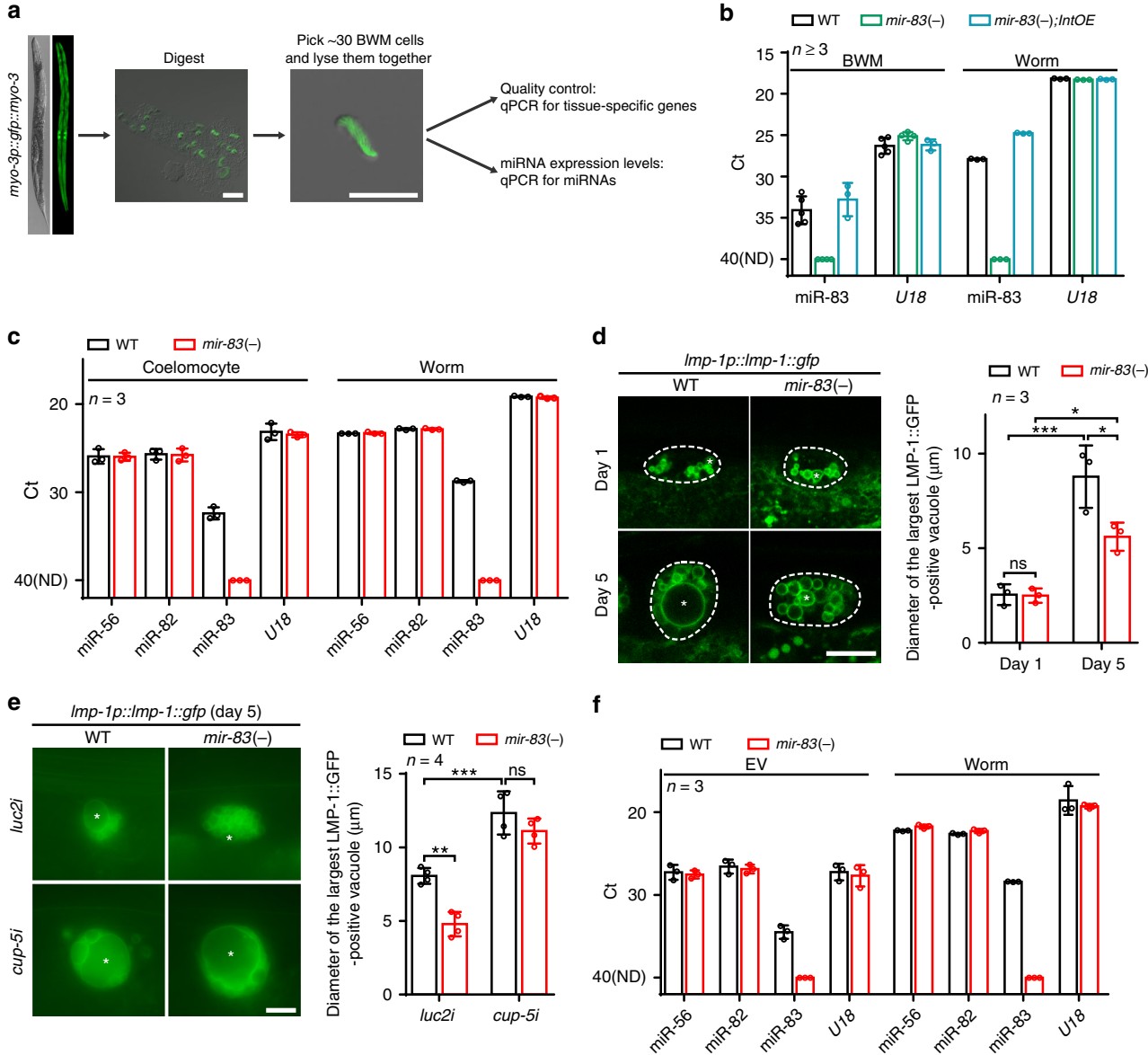

**Fig. 6** *mir-83* is transported from the intestine to the body wall muscle potentially through the extracellular vesicles in the pseudocoelomic fluidics. **a** The workflow of the single cell qPCR from the isolated BWM cells. Scale bars: 50 μm. **b** qRT-PCR of miR-83 and U18 in the isolated BWM cells and whole worms of indicated strains. Cells were collected from worms at day 1 of adulthood. Note that miR-83 is detected in the BWM cells of WT worms and the *mir-83(-)* mutants with an intestine specific transgene expressing *mir-83* (*mir-83(-);IntOE*). Note that Ct anti-correlates with the expression level. Samples without qRT-PCR signal were set as Ct 40. ND: not detected. n = 5, 4, and 3 independent experiments for WT, *mir-83(-)*, and *mir-83(-);IntOE*, respectively. **c** miR-83 is detected in the isolated coelomocytes by single cell qRT-PCR. Cells were isolated from worms at day 1 of adulthood. n = 3 independent experiments. **d** *mir-83* mutation reduces the increased size of LMP-1::GFP-positive vacuoles in the aged coelomocytes. The largest vacuoles (asterisks) were measured. Scale bar: 10 μm. n = 3 independent experiments containing at least 26 worms. **e** The size of the largest LMP-1::GFP-positive vacuoles (asterisks) in the aged coelomocytes of WT worms and *mir-83(-)* mutants upon indicated RNAi treatments. Scale bar: 10 μm. n = 4 independent experiments containing at least 60 worms. **f** qRT-PCR of indicated genes in the purified extracellular vesicles (EVs) and worms of the indicated strains. n = 3 independent experiments. Statistical significance was calculated by Two-way ANOVA. ns: non-significant, *p < 0.05, **p < 0.01, ***p < 0.001. Source data are provided as a Source Data file

Fig. 1c-d)[16,17], it was detected in isolated coelomocytes by single-cell qPCR (Fig. 6c and Supplementary Fig. 5g-h). We further examined LMP-1-labeled lysosomes in coelomocytes and found that they turned into large vacuoles at day 5 of adulthood, reminiscent of the phenotypes observed upon inhibition of *cup-5*[22] (Fig. 6d). As in the intestine, mutation of *mir-83* suppressed the enlargement of LMP-1-positive vacuoles in the aged coelomocytes, and *cup-5* RNAi restored the sizes of LMP-1-positive vacuoles in the coelomocytes of *mir-83(-)* mutants to WT levels (Fig. 6d, e). Hence, *mir-83* is also present in coelomocytes and could control lysosomes by inhibiting *cup-5*.

Previous studies have suggested that extracellular vesicles (EVs) transfer microRNAs among cells[34]. Worm EVs, mostly exosomes given their size, were isolated to determine whether *mir-83* is also transferred in EVs (Supplementary Fig. 6a-c). *mir-56* and *mir-82*, two microRNAs proposed to be enriched in EVs[35], were abundant in our isolated EVs (Fig. 6f), further validating the EV preparation. As expected, *mir-83* was detected

in the isolated EVs from WT worms but not in those from *mir-83 (-)* mutants (Fig. 6f and Supplementary Fig. 6d). These data therefore imply that *mir-83* could be shuttled into BWM by EVs.

**mir-83 regulates ageing by inhibiting autophagy and cup-5.** Autophagy is crucial for protein homeostasis and longevity[36]. Consistent with the effects of *mir-83* on autophagy, *mir-83* deletion reduced the numbers of polyQ aggregates in the intestines and BWM of worms at days 5 and 3 of adulthood, respectively (Fig. 7a, b), indicating the importance of *mir-83* in protein homeostasis. Consistent with the role of *mir-83* in protein homeostasis, mutation of *mir-83* extended the lifespan of WT animals (Fig. 7c; Supplementary Data 1), as recently reported[37]. *mir-83(-)* mutants also exhibited a mild egg-laying defect (Supplementary Fig. 7), suggesting that this microRNA functions in reproduction. Inhibiting *cup-5* by RNAi abolished the decrease in polyQ aggregates and the extension of lifespan in *mir-83(-)* mutants (Fig. 7d–f; Supplementary Data 1), further indicating that *mir-83* promotes ageing by inhibiting *cup-5* and dysregulating autophagy. Accordingly, disruption of autophagy with chloroquine reduced the lifespan of *mir-83(-)* mutants to that of WT worms (Fig. 7g; Supplementary Data 1). As ageing upregulated *mir-83* in the intestine (Fig. 1c, d and Supplementary Fig. 1c-d), intestinal overexpression of *mir-83* blocked the longevity of *mir-83(-)* mutants (Fig. 7h). Taken together, these data indicate that the ageing-related upregulation of *mir-83* in the intestine impairs protein homeostasis and longevity by disrupting autophagy.

## Discussion

Autophagy is gradually impaired with ageing, resulting in the disruption of proteostasis[3]. As ageing progresses, autophagy systematically declines in distinct tissues in *C. elegans*[10]. However, the machinery co-ordinately decreasing autophagy across tissues with age has been poorly studied. In this study, we identified a secreted microRNA, *mir-83*/miR-29, as an age-controlled messenger that dysregulates autophagy among tissues by directly inhibiting *cup-5*/MCOLN (Fig. 7i).

Autophagy is generally a three-step process that involves sequestration of cytoplasmic components, degradation, and reuse of the degraded products[2]. A target of *mir-83*, *cup-5*, is a key regulator in these steps. *cup-5* was first recognized for its essential role in the degradation step of autophagy[22], and *mir-83* indeed suppresses the degradation of autophagy substrates through *cup-5*. The mammalian homolog of *cup-5*, MCOLN, was also found to be an important regulator of TFEB, a master transcription factor controlling multiple steps of autophagy[26,27]. A recent report suggested *hlh-30*/TFEB is under the regulation of *mir-83*[37]. Our results showed that *mir-83* inhibited GFP::LGG-1 expression through *cup-5*, implying that the interaction between *cup-5*/MCOLN and *hlh-30*/TFEB could be well conserved in evolution and indicating that *mir-83* also disrupts autophagy by affecting its initiation. Consistently, *mir-83* reduced the numbers of APs and ALs through *cup-5*. Therefore, *mir-83* impairs multiple steps of autophagy by inhibiting CUP-5.

With ageing, autophagy is systematically dysregulated in distinct tissues in *C. elegans*[10], implying the existence of inter-tissue crosstalk regulating autophagy. Secreted microRNAs have been proposed to be potential messengers in inter-tissue signaling[38]. After using two independent transcriptional reporters, we and two other labs report that *mir-83* is not transcribed in BWM[16,17]. qPCR of isolated BWM cells confirmed that the *mir-83* promoter is inactive in BWM cells. However, *mir-83* is detectable in isolated BWM cells, indicating that it is a secreted microRNA. We cannot exclude the possibility that a trace amount of *mir-83* below the

detection limit may still be transcribed in BWM, as suggested by a recent report[37]; however, *mir-83* expressed specifically in the intestine, but not *gfp* driven by the same intestine-specific promoter, was also detected in BWM. Together with the detection of *mir-83* in EVs, this finding indicates that *mir-83* is a secreted microRNA. Its translocation from the intestine into BWM impairs autophagy in the receiving tissue by suppressing CUP-5. Our results therefore reveal a mechanism of inter-tissue regulation of autophagy and identify *mir-83*, a secreted microRNA, as its modulator. In worms, several neurotransmitters and a Wnt ligand have been identified as signals that are secreted from the nervous system to control the unfolded protein response, another essential element of proteostasis, in other tissues[7,8]. Our results not only show that a secreted microRNA is an inter-tissue messenger controlling autophagy for proteostasis but also indicate that tissues other than nervous system (e.g., the intestine) broadcast signals for proteostasis throughout the body. Similarly, transcellular chaperone signaling from muscle to intestine and neurons is important in the response against proteotoxic stress[5,6].

*mir-83* was present in purified EVs and coelomocytes. Given this finding and the changes in lysosome morphology in coelomocytes, which were due to the interaction of *mir-83* and *cup-5*, we speculate that the transportation of *mir-83* from the intestine to BWM potentially occurs through EVs in pseudocoelom fluids, the body fluids of worms. Coelomocytes, which are active in endocytosis, may scavenge *mir-83* in pseudocoelom fluids and thereby control its transportation across tissues. In addition, other microRNAs were also detected in the coelomocytes and EVs secreted by worms in our study, suggesting that inter-tissue transportation is not specific for *mir-83*. In mammals, secreted microRNAs are also abundant in body fluids and are linked to ageing[13]. Our results suggest that molecules in this class are not only ageing biomarkers but could also function as key messengers in regulating ageing progression (e.g., the disruption of autophagy) in distinct tissues. It will be intriguing to study in the future how microRNAs are transported across tissues.

The levels of pri-miR-83 are much lower than those of mature miR-83 in worms, potentially leading to the transport of many more mature miR-83 molecules than pri-miR-83 molecules. This could be why we did not detect pri-miR-83 in isolated BWM cells, coelomocytes, or EVs. Nevertheless, the precursors of *mir-83* (pri- and pre-miR-83) could still be transported across tissues in addition to the mature form. Future studies on the detailed species of *mir-83* undergoing inter-cellular transportation could help elucidate the mechanism of microRNA transport across tissues.

Evolution is likely to enrich alleles crucial to youth fitness while neglecting the detrimental effects of these alleles after reproduction due to declining natural selection pressure with ageing[39]. Consistent with previous RNA-Seq studies[40], our study presented multiple lines of evidence that *mir-83* is upregulated in aged worms without drug treatment, whereas Dzakah et al. observed decreases in *mir-83* in FUdR-treated worms[37]. In untreated worms, the upregulation of the *hsf-1-mir-83* axis with ageing could be a good example of the antagonistic pleiotropy hypothesis. *mir-83* contributes to the development of the germline in larvae and to egg-laying in reproductive adults[17], but its upregulation by HSF-1 in aged worms impairs autophagy in multiple tissues. *hsf-1*/HSF1 is a multi-faceted transcription factor for cell survival and growth. It is a master regulator that promotes autophagy and protein homeostasis, especially under stresses[41], whereas its inactivation causes small cell/body sizes and infertility[42,43]. In the first 10 days of adulthood, *C. elegans* grow by 70% in length solely by increasing cell size[44]. The increase in HSF-1 in aged worms might be a response to this change. However, this increase induces detrimental pleiotropic side effects

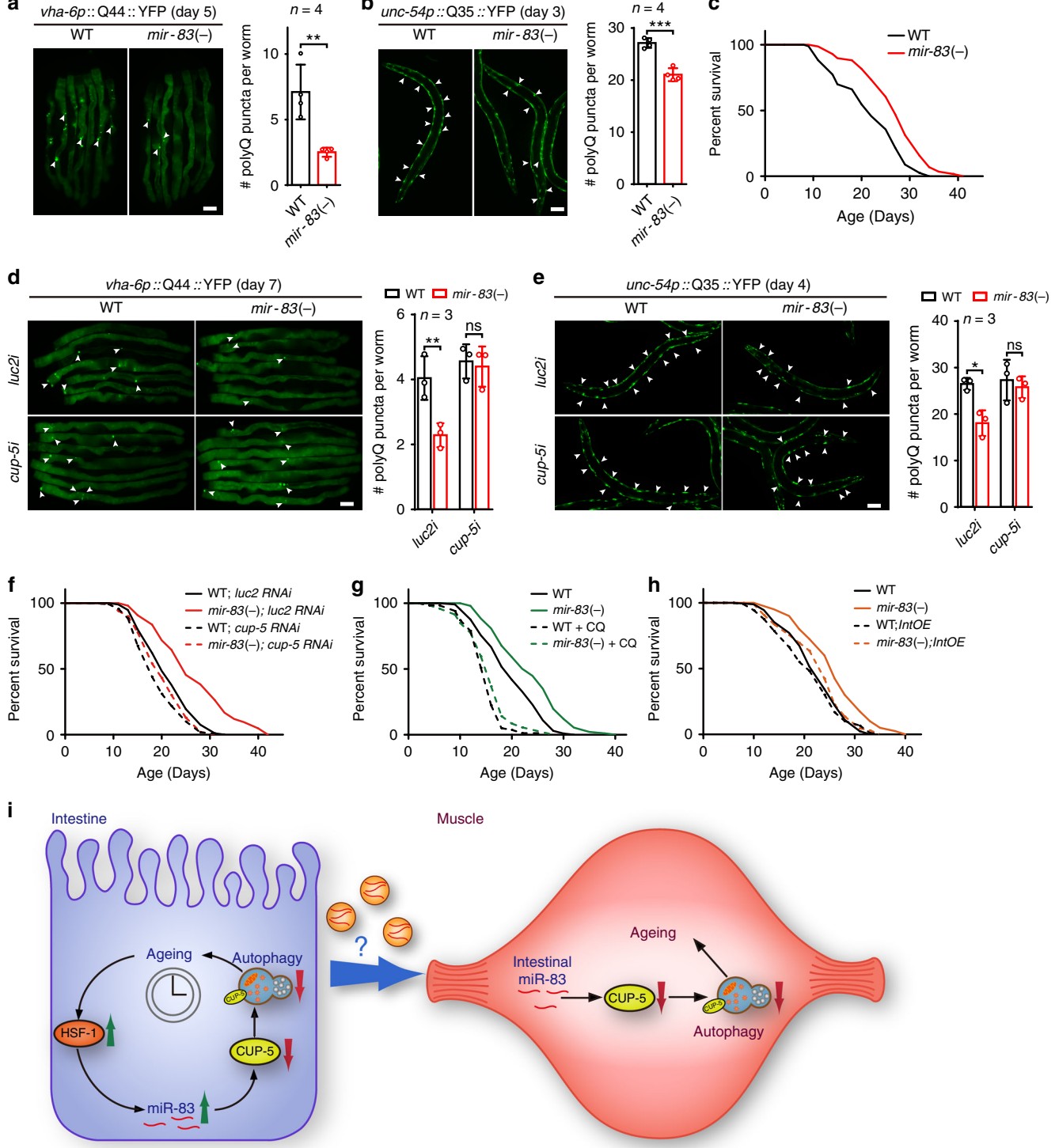

**Fig. 7** *mir-83* modulates ageing through *cup-5* and autophagy. **a, b** PolyQ-YFP aggregates (arrow heads) in the intestine (**a**) and BWM (**b**) of WT worms and *mir-83(-)* mutants at indicated ages. n = 4 independent experiments containing at least 79 worms. **c** Survival curves of indicated strains. **d, e** RNAi against *cup-5* increases the polyQ aggregates in the intestine (**d**) and BWM (**e**) of *mir-83(-)* mutants to WT levels. n = 3 independent experiments containing at least 44 worms. **f** *cup-5* RNAi abolishes the extended lifespan of *mir-83(-)* mutants. **g** Blocking autophagy with 5 mM chloroquine (CQ) abolishes the extended lifespan of *mir-83(-)* mutants. **h** Ageing assays of the indicated strains. *IntOE*: an intestine-specific transgene expressing *mir-83*. **i** *mir-83* is upregulated in the aged intestine by increased HSF-1 and transported across tissues to disrupt autophagy through inhibiting CUP-5. See Discussion for details. Scale bar: 100 μm. Statistical significance was calculated by unpaired *t*-test in **a, b** or Two-way ANOVA in **d, e**. ns: non-significant, *$p < 0.05$, **$p < 0.01$, ***$p < 0.001$. See Supplementary Data 1 for lifespan statistics (**c** and **f**–**h**). Source data are provided as a Source Data file

of *mir-83* upregulation and subsequent impairment of autophagy and proteostasis. In young worms, *hsf-1* upregulates autophagy following hormetic heat stress[41]. Our results therefore suggest that *hsf-1* plays a dual role in autophagy, promoting autophagy in young worms upon heat stress and disrupting it in aged animals. Consistent with this speculation, heat shock upregulates multiple autophagy genes in young worms but not in aged animals[41]. Similar to our findings, inactivation of *pha-4*/FOXA, a key transcription factor in dietary restriction-induced longevity, in worms post reproduction actually extends lifespan[45].

In addition to the increase in HSF-1 in aged worms, epigenetic factors may also contribute to the late activation of the *hsf-1-mir-83* axis in aged worms. Trimethylation of Lys 27 on H3 (H3K27me3) has been reported to interfere with HSF-1 binding to genomic DNA and to suppress the transcription of HSF-1 targets[46]. The transcription of *mir-83* could be under similar control, since ChIP-Seq results indicate strong binding of H3K27me3 and another histone modification anti-correlated with gene expression (H3K9me3) at the promoter of *mir-83* in young adults and larvae[47]. Because histone modifications undergo dramatic changes with ageing and have remarkable variation across tissues[48,49], the specific spatial and temporal pattern of histone modification at the *mir-83* promoter may thus suppress the response of *mir-83* to *hsf-1* in young animals but facilitate the upregulation of *mir-83* by *hsf-1* in intestines of aged animals.

In summary, we have identified *mir-83*/miR-29 as an ageing-induced microRNA messenger from the intestine that systematically disrupts autophagy in multiple tissues by inhibiting *cup-5*/MCOLN. *mir-83* is well conserved in evolution, and its mammalian orthologue, miR-29, is upregulated in the brains and the serum of aged patients with hypertrophic cardiomyopathy[50,51]. In addition, although the 3′-UTR of MCOLN does not harbor miR-29 binding sites, miR-29 is predicted to target a few other autophagy genes, including ATG4D, the mammal orthologue of *atg-4.2*[52]. Therefore, it will be interesting to explore the function of miR-29 as an inter-tissue messenger in impaired autophagy in aged individuals. In addition, as miR-29 is reported to interfere with biological processes other than autophagy, it will also be interesting to explore the other functions of secreted *mir-83* in worms.

## Methods

***C. elegans* strains and culture**. *C. elegans* strains used in this study are listed in Supplementary Data 3. *C. elegans* were grown with standard techniques at 20 °C, unless otherwise noted[53]. Some strains were provided by the CGC, which is funded by NIH Office of Research Infrastructure Programs (P40 OD010440). The transgenic lines of HZ589 and bpIs193 were kindly provided by Hong Zhang lab[54], and the transgenic lines expressing *dFP::lgg-1* (DLM3 and DLM11) were gifts from Miller lab[30].

To collect worms at different ages, worms were synchronized by collecting eggs and transferred to 60 mm plates (20 worms per plate). Worms were transferred to fresh plates every other day until day 10 to avoid the contamination from progenies, and subsequently monitored every other day for dead worms. Worms undergoing internal hatching, bursting vulva, crawling off the plates, or contamination as well as dead worms were removed from the plates.

**Cell culture**. HEK293T cells (ATCC) were maintained in DMEM medium supplemented with 10% fetal bovine serum (Gibco) at 37 °C, 5% $CO_2$.

**Lifespan assays**. Adult lifespan analysis was performed as reported[24]. For lifespan assays under chloroquine treatment, worms were incubated on NG plates containing 5 mM chloroquine diphosphate salt (Sigma-Aldrich) from eggs. During reproductive period, worms were handpicked onto fresh plates every other day to avoid progeny contamination. Worms not responding to prodding were scored as dead. Worms undergoing internal hatching, bursting vulva, crawling off the plates, or contamination were censored. Statistical analysis was performed with the Mantel-Cox Log Rank method.

**Plasmid construction**. For luciferase reporters with 3′-UTR, 3′-UTRs of interest were amplified from N2 genomic DNA by PCR and cloned after the luciferase gene

in pGL3-M (a gift from Zhu Lab). To generate pGL3-M-*cup-5u*M and pGL3-M-*atg-4.2u*M, the *mir-83* binding sites (5′-TGGTGCT-3′) in the 3′-UTRs were mutated to 5′-ACCACGA-3′ by PCR.

To generate *mir-83p::gfp::mir-83*, 3110 bp of *mir-83* promoter and 500 bp of *mir-83* stem loop with flanking sequence were amplified from N2 genomic DNA and cloned into the 5′ and 3′ region of *gfp* in L3781, respectively. The Fire Lab *C. elegans* Vector Kit was a gift from Andrew Fire.

To generate *cup-5p::mCherry::cup-5u*, gfp in L3781 was replaced by mCherry, 1.7 kb of *cup-5* promoter and 229 bp of *cup-5* 3′-UTR were amplified from N2 genomic DNA and cloned into L3781-mCherry.

To generate *myo-3p::mCherry::cup-5u*, 2,385 bp of *myo-3* promoter and 229 bp of *cup-5*-3′UTR were amplified from N2 genome, and cloned into L3781-mCherry.

To generate *atg-4.2p::mCherry::atg-4.2 u*, 2 kb of *atg-4.2* promoter and 101 bp of *atg-4.2* 3′-UTR were amplified from N2 genomic DNA and cloned into L3781-mCherry.

To generate *cup-5p*::mCherry::*cup-5*cDNA::*cup-5u*, 1.8 kb of *cup-5* cDNA was amplified from N2 cDNA and cloned between mCherry and *cup-5* 3′-UTR of *cup-5p*::mCherry::*cup-5u*.

To generate *ges-1p::mir-83*, L3781 was first linearized by PCR to remove the gfp within. 2.2 kb of *ges-1* promoter, 500 bp of *mir-83* stem loop with flanking sequence were amplified from N2 genomic DNA and cloned into the linearized L3781.

To generated *ges-1p::gfp::mir-83*, 2.2 kb of *ges-1* promoter, 500 bp of *mir-83* stem loop with flanking sequence were amplified from N2 genomic DNA and cloned into L3781.

To generate *mir-83p::TagRFP-T* and *ges-1p::TagRFP-T*, L3781 was first linearized by PCR to remove the *gfp* within. TagRFP-T was amplified from pDD284 (a gift from Bob Goldstein, Addgene # 66825) and cloned into the linearized L3781. 2.2 kb of *ges-1* promoter and 3,110 bp of *mir-83* promoter were amplified from N2 genomic DNA and cloned into the 5′ region of TagRFP-T respectively.

For genome editing by CRISPR-Cas9, plasmids were constructed as described[55]. To generate pDD284-*myo-3p*::RFP::3xFLAG::*cup-5u*, the 5′ and 3′ homologous arms of universal MosSCI site Uni_II, 2,385 bp of *myo-3* promoter and 229 bp of *cup-5* 3′-UTR were amplified from N2 genome. The 5′ arm of Uni_II was inserted into pCR-BluntII-TOPO vector (Sigma-Aldrich), and was mutated at the PAM (NGG motif) by PCR. UniII 5′M-*myo-3p* and *cup-5u*-UniII 3′ fragments were amplified from the subclones of pCR-BluntII-TOPO-UniII 5′M-*myo-3p* and pCR-BluntII-TOPO-*cup-5u*-UniII 3′ and inserted into pDD284 (a gift from Bob Goldstein, Addgene # 66825) using NEBuilder HiFi DNA Assembly Master Mix (New England Biolabs).

To generate pDD282-*hsf-1::gfp::3 × FLAG::Avi*, Avi was inserted at the 3′ terminal of 3xFLAG in pDD282 (a gift from Bob Goldstein, Addgene # 66823). 5′ and 3′ homologous arms of *hsf-1* were amplified from N2 genome and inserted into pDD282-*Avi* using NEBuilder HiFi DNA Assembly Master Mix (New England Biolabs).

To generate pDD162-UniII and pDD162-*hsf-1*, the sgRNA sequences targeting Uni_II (5′-CCACTTTCAATGTGAGTTGT-3′) and *hsf-1* C-terminus (5′-AAGTCCATCGGATCCTAATT-3′) were respectively designed using Zhang lab's CRISPR design tool (http://crispr.mit.edu/) and cloned into pDD162 (a gift from Bob Goldstein, Addgene # 47549) as instructed[55].

**Transgenes**. Plasmids *cup-5p::mCherry::cup-5u* (50 ng/µl), *atg-4.2p::mCherry::atg-4.2 u* (50 ng/µl) and *cup-5p::mCherry::cup-5cDNA::cup-5u* (50 ng/µl) were injected into N2 to generate sydEx024, sydEx026 and sydEx164 respectively with an injection marker of *sur-5p::gfp* (50 ng/µl).

Plasmids *ges-1p::gfp::mir-83* (50 ng/µl), *mir-83p::gfp::mir-83* (50 ng/µl), and *ges-1p::mir-83* (50 ng/µl) were injected into N2 to generate sydEx078, sydEx080, and sydEx121 respectively with an injection marker of *myo-2::mCherry* (2.5 ng/µl).

Plasmid of *myo-3p::mCherry::cup-5u* (50 ng/µl) was injected into N2 to generate sydEx086 with an injection marker of L4816 (50 ng/µl).

To generate sydIs073, plasmids of pDD162-UniII (50 ng/µl) and pDD284-*myo-3p*::RFP::3xFLAG::*cup-5u* (10 ng/µl) were coinjected into N2 with an injection marker of *myo-2p::mCherry* (2.5 ng/µl). Screening of integrated strain is as reported[55].

To generate *hsf-1(syd101[hsf-1::gfp::3 × FLAG::Avi])* I, plasmids of pDD162-*hsf-1* (50 ng/µl) and pDD282-*hsf-1::gfp::3 × FLAG::Avi* (10 ng/µl) were coinjected into N2 with an injection marker of *myo-2p::mCherry* (2.5 ng/µl). Screening of integrated strain is as reported[55].

To generate sydEx143, the plasmid of *coel::rfp* (50 ng/µl) was injected into N2. The *cup-5(syb1028)*III allele was generated by SunyBiotech using CRISPR/Cas9 technology. 3xFLAG and wrmScarlet (789 bp) were insert into the N terminal of the endogenous *cup-5* gene. The strain was verified by DNA sequencing.

To generate sydEx198[*ges-1p::TagRFP-T*] and sydEx200[*mir-83p::TagRFP-T*], the plasmids of *ges-1p::TagRFP-T* and *mir-83p::TagRFP-T* were injected into RW1596 respectively.

**RNA interference**. RNAi experiments were based on a reported protocol[56]. In brief, worms were grown on HT115 expressing dsRNA against indicated genes from egg until corresponding time. HT115 expression dsRNA against *luc2* (a real gene but irrelevant to worm biology) served as a control. The strains of HT115

[L4440::luc2] and HT115 [L4440::cup-5] are gifts from Adam Antebi lab in Max Planck Institute for Biology of Ageing. The strain of HT115 [L4440::hsf-1] is a gift from Shiqing Cai lab in Institute of Neuroscience, CAS.

**Cell transfection and luciferase assay**. HEK293T cells were seeded into a 24-well plate and transfected using Lipofectamine 3000 Transfection Reagent (Thermo Fisher Scientific) following the manufacturer's instruction one day post seeding. When testing the interaction between cel-mir-83-3p and 3′-UTRs of interest in HEK293T cells, 14 pmol of microRNA mimic, 50 ng of the 3′-UTR reporter plasmid, and 5 ng of Renilla luciferase plasmid were added into each well. MicroRNA mimics were ordered from GenePharma. Sequences of these mimics are listed in Supplementary Data 4.

Cells were collected and examined for luciferase activity 48 h post transfection by Dual-Luciferase® Reporter Assay System (Promega) and Synergy™NEO (Bio Tek) as the manufacturer instructed. At least three independent experiments were performed.

**Isolation of worm cells**. Worms were lysed following a modified protocol from Murphy lab[57]. In brief, about 2000 synchronized worms of day 1 adulthood were washed by M9 to remove excess bacteria. Pelleted worms were incubated in 1 ml lysis buffer (200 mM DTT, 0.25% SDS, 20 mM HEPES, 3% sucrose) with rocking at 700 rpm for 7 min at 20 °C. The worms were washed by 1 ml PBS for 6 times to remove the lysis buffer. The worm pellet was incubated in 400 μl Pronase (20 mg/ml) with rocking at 1200 rpm for 10 min at 20 °C. The proteinase was removed by washing worms with 1 ml PBS for 6 times.

To isolate coelomocytes, ~100 μl digested worms were placed on a glass slide. On an Olympus IX73 microscope, coelomocytes, identified by unc-122p::DsRed, were picked using a broken microinjection needle (Harvard Apparatus) and transferred into 200 μl PBS with 0.1% BSA. Cells were further transferred with the needle into ~4 μl PBS with 0.1% BSA. About 20 coelomocytes were collected for each sample.

To isolate BWM cells, worms were further digested in 400 μl 0.2% collagenase IV with rocking at 1200 rpm for about 40 min at 20 °C. When most worm bodies were dissociated and lots of single BWM cells were identified by GFP::MYO-3, the collagenase was removed by centrifugation. Pelleted cells and worm debris were resuspended in 1 ml PBS with 0.1% BSA. BWM cells were picked as described above. About 30 BWM cells were collected for each sample.

**Quantitative RT-PCR**. qRT-PCR was performed following a previous report[24]. In brief, for mRNA, total RNA was prepared by RNeasy Mini kit (QIAGEN) from ~100 worms. cDNA was subsequently generated by iScript™ Reverse Transcription Supermix for RT-qPCR (Bio-Rad). For microRNA, miRNeasy Mini kit (QIAGEN) and TaqMan MicroRNA Reverse Transcription Kit (Thermo Fisher Scientific) were used for total RNA and cDNA preparation, respectively. qRT-PCR was performed with Bestar® Sybr Green qPCR Master Mix (DBI Bioscience) or 2xNovoStart®SYBR qPCR SuperMix Plus (Novoprotein) on a QuantStudio™ 6 Flex Real-time PCR System (Applied Biosystems) or a CFX384 Touch™ Real-Time PCR Detection System (Bio-Rad). Four technical replicates were performed in each reaction. A combination of ama-1, act-1 and cdc-42 was used as reference for mRNA quantification, whereas Sno-RNA U18 as a reference for microRNA qRT-PCR. The results were from at least three biological replicates. qRT-PCR primers for mir-83 was designed as reported[58,59]. The sequences of all reported and self-designed primers are listed in Supplementary Data 4[24,60,61].

For qRT-PCR of singled cells, isolated cells were first collected in ~4 μl PBS with 0.1% BSA and lysed at room temperature for 5 min using 6 μl of lysis buffer (miScript Single Cell qPCR Kit, QIAGEN). Cell lysates were first incubated at 75 °C for 5 min to inactivate DNase and then divided into two sets of samples respectively for mRNA and microRNA analysis.

For mRNA analysis, cDNA was prepared as reported[62]. In brief, the 1 μl 3′CDS primer (10 μM) (Sango Biotech Biotech) and 1 μl dNTPs (10 mM) was added into 1–5 μl cell lysate and incubated at 72 °C for 3 min. The cell lysate was subsequently mixed with 0.5 μl PrimeScript RT Enzyme Mix I (Takara), 0.25 μl RNAse inhibitor (40 U/μl) (Thermo Fisher Scientific), 2 μl 5× PrimeScript Buffer (Takara), 0.5 μl DTT (100 mM), 2 μl Betaine (5 M), 0.06 μl MgCl2 (1 M), 0.1 μl LNA-TSO primer (100 μM) (Sango Biotech) and 1.6 μl nuclease-free water. The mixture was incubated at 37 °C for 90 min, then subjected to 15 cycles of reverse transcription (50 °C for 2 min, 37 °C for 2 min), and finally incubated at 85 °C for 5 s for reverse transcription. The cDNA product was mixed with 0.25 μl Q5 High-Fidelity DNA Polymerase (New England Biolabs), 5 μl 5× Q5 Reaction Buffer (New England Biolabs), 0.5 μl dNTPs (10 mM), 1.25 μl A-IS PCR primer (Sango Biotech) and 8 μl nuclease-free water, and then amplified using the following thermocycler settings: 98 °C for 30 s, 25 cycles of amplification (98 °C for 10 s, 67 °C for 30 s and 72 °C for 6 min), 72 °C for 5 min. The amplified cDNA was diluted by 10 times for qRT-PCR.

The microRNAs were reverse transcribed using TaqMan MicroRNA Reverse Transcription Kit (Thermo Fisher Scientific). In brief, ~3.5 μl cell lysate were mixed with 0.4 μl RT primer of microRNAs (4 μM), 0.4 μl U18 reverse primer (4 μM), 0.4 μl dNTPs (100 mM), 3 μl MultiScribe™ Reverse Transcriptase (50 U/μl), 1.6 μl 10× RT buffer, 1.8 μl MgCl2 (25 mM), 0.2 μl RNase Inhibitor (20 U/μl), and 1.4 μl

nuclease-free water. The mixture was incubated on ice for 5 min, subjected to 40 cycles of reverse transcription (16 °C for 2 min, 42 °C for 1 min, and 50 °C for 1 s), incubated at 85 °C for 5 min, and kept at 4 °C. 1 μl of reverse transcribed sample was used as the template in each well of a 384-well qPCR reaction.

**RNA-Seq**. In each one of the three biological replicates, ~300 N2 worms at indicated ages were handpicked into TRIzol (Invitrogen) for each time point. Total RNA isolation was performed using QIAGEN miRNeasy Mini (QIAGEN) following the manufacturer's instruction. Small RNA library preparation and sequencing were performed with Illumina sequencing technology by BGI, as reported[63]. The reads were first aligned to the C. elegans reference genome (WBcel215.67) using Tophat[64]. Aligned reads were then mapped against known microRNA sequences by miRDeep[65]. Differentially expressed microRNAs were figured out using the R package DESeq[66].

**Microscopy**. For microscopy experiments, worms were mounted on 5% agar pads and anesthetized using 5 mM levamisole for imaging LGG-1 puncta, or 0.1% sodium azide for the rest assays. The images were captured using a Leica TCS SP8 or an Olympus BX53 microscope. Fluorescence intensity were measured by Adobe Photoshop. Background signal was subtracted as reported[24].

**Autophagy analysis**. GFP::LGG-1 puncta of posterior intestine cells were counted from about 5 slices with 1 μm step size, the Z-position was selected where intestinal nucleus could be seen clearly. GFP::LGG-1 puncta of BWM cells were counted from 1 slice where the striation could be clearly seen. The puncta were counted using ComDet v.0.3.7 in ImageJ. For each genotype, at least 22 animals at corresponding stages from three independent experiments were scored.

Images of mCherry::GFP::LGG-1 puncta were scored using the same method as examining GFP::LGG-1 puncta. The mCherry::GFP::LGG-1 puncta of head neurons were counted from 1 slice where most neurons surrounding the posterior pharynx bulb are clearly seen. The number of APs was calculated as the GFP-positive puncta, and the number of ALs was calculated as the difference of the mCherry-positive puncta and the mCherry-GFP-positive puncta. For each genotype, at least 16 animals at corresponding stages from at least three independent experiments were scored.

To capture SQST-1::GFP images, head neurons and BWM cells were identified by morphology using differential interference contrast microscopy. The head neurons between two bulbs of pharynx and the BWM cells near the tail of worm were examined. For each genotype, at least 45 animals at corresponding stages from at least three independent experiments were scored.

Worms were transferred onto NGM plates containing 5 mM chloroquine and incubated for 4 h or 1 h to block the autophagy flux in the neurons or in other tissues.

**Analysis of lysosome number and morphology**. Lysosome numbers of anterior intestine cells were counted from about 5 slices with 1 μm step size, the Z-position was selected where intestinal nucleus could be clearly seen. For each genotype, at least 20 animals at corresponding stages from at least three independent experiments were scored.

The images of lysosome morphology in coelomocytes were captured from the pair near worm tail. The diameters of large LMP-1::GFP positive vacuole of WT worms and mir-83(-) mutants were quantified by LAS X. The diameters of large LMP-1::GFP positive vacuole of worms under RNAi treatment were quantified by ImageJ. For each genotype, at least 45 animals at corresponding stages from three independent experiments were scored.

**Analysis of polyQ strains**. The numbers of intestinal and muscular polyQ aggregates in WT worms and mir-83(-) mutants were counted in individual worms at indicated days of adulthood. For each genotype, at least 45 animals at corresponding stages from three independent experiments were scored.

**Western blotting**. Synchronized worms at corresponding stages of adulthood were harvested in M9. After three rounds of freezing and thaw, worms were lysed by adding 4x loading buffer. Proteins were separated by reducing SDS-PAGE and transferred to PVDF membranes. Membranes were then blotted with primary antibodies against GFP (1:200 dilution, Cat# sc-9996, Santa Cruz), FLAG (1:2000 dilution, Cat# F7425, Sigma-Aldrich), α-tubulin (1:2000 dilution, Cat# T5168, Sigma-Aldrich), or mCherry (1:2,000 dilution, Cat# NBP1-96752, Novus Biologicals), and secondary antibodies against Rabbit IgG (1:5000 dilution, Cat# G-21234, Thermo Fisher Scientific) or Mouse IgG (1:5000 dilution, Cat# G-21040, Thermo Fisher Scientific). To harvest worms at day 5 and day 7 of adulthood, each sample was prepared from about 120 synchronized L4 worms on six 60 mm plates. The worms were transferred to fresh plates every two days until reaching desired ages. Worms undergoing internal hatching and vulva bursting were discarded. Signals of western blotting were measured using Adobe Photoshop. Background signals were subtracted as reported[24]. All the uncropped and unprocessed images were shown in the Supplementary Fig. 8.

**Isolation of extracellular vesicles**. Synchronized worms were grown in liquid culture with the density of 3 worms per microliter and fed with 25 mg OP50 per milliliter until day 1 of adulthood[67]. Because a proportion of extracellular vesicles are secreted out of the worms[68–70], the synchronized worms were soaked in M9 to collect EVs. In brief, for each sample, 300,000 worms were collected and washed by M9 to remove excess bacteria. Worms were subsequently cultured in 25 ml M9 at 20 °C for 12 h with shaking at 180 rpm. Worms were sedimented by gravity, whereas the supernatant was collected and filtered by a 0.22 μm Millex-GP Syringe Filter Unit (Millipore). Extracellular vesicles (EVs) were isolated from the filtered medium using Total Exosome Isolation Reagent (Thermo Fisher Scientific) following the manufacturer's instruction. For gene expression, the isolated EVs were suspended in 700 μl QIAzol Lysis Reagent (QIAGEN). EVs were suspended in 600 μl PBS for other experiments.

**Transmission electron microscopy of isolated extracellular vesicles**. In total 10–20 μl EVs suspended in PBS were deposited onto a carbon coated electron microscopy grid, stained with 2% uranyl acetate, and examined using a Tecnai G2 Spirit microscope (Thermo Fisher Scientific).

**Nanoparticle tracking of isolated extracellular vesicles**. EVs in PBS were diluted 100-fold in PBS and subjected to nanoparticle tracking using a NanoSight NS300 (Malvern Instruments).

**Statistical analysis**. Results are presented as Mean ± SD unless otherwise noted. Statistical tests were performed as indicated using GraphPad Prism (GraphPad software). The statistics of Poisson regression was performed with R Core Team as reported[10]. Detailed statistics for lifespan assays and other experiments were respectively listed in Supplementary Data 1 and 2.

**Reporting summary**. Further information on research design is available in the Nature Research Reporting Summary linked to this article.

## Data availability

The source data underlying all figures are provided as a Source Data file. The associated data are also available at figshare [https://figshare.com/, https://doi.org/10.6084/m9.figshare.8875115]. The RNA-Seq data in this publication are accessible through GEO Series accession number GSE136612. All other relevant data are available from the authors.

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

## Acknowledgements

We thank Dr. Xueliang Zhu (SIBCB, CAS) for plasmids, Dr. Jorge Boucas, Dr. Franziska Metge and the bioinformatic core (MPI-AGE) for RNA-Seq data, the CGC, Dr. Hong Zhang (IBP, CAS), and Dr. Dana L Miller (University of Washington) for strains, and Mr. Chenghui Wan for illustration. This research was supported by the Thousand Talents Plan (Youth), the Strategic Priority Research Program of the Chinese Academy of Sciences, Grant No. XDB19000000, NSFC, Grant No. 31571412, SYBACOL/BMBF, Grant No. FKZ0315893B, and Max Planck Gesellschaft.

## Author contributions

Y.Z. and Y.S. conceived the project, designed and performed the experiments, and analyzed data. M.S. performed the quantification of microscopy images and contributed to transgenic strain preparation and worm culture. X.W. performed the extracellular vesicles-related analysis. Z.H. constructed the strain of HSF-1::GFP::3xFLAG and contributed to the analysis of its expression with ageing. G.C., G.P., and N.J. helped in single cell qPCR. C.D. and A.A. helped with RNA-Seq data analysis. Y.Z. and Y.S. wrote the manuscript. M.S., X.W., and N.J. contributed to manuscript editing.

## Competing interests

The authors declare no competing interests.
