## [Peer Review File · Nature Communications]

Reviewers' comments:

Reviewer #1 (Remarks to the Author):

In this interesting manuscript, Zhou et al. study the role of microRNAs in dysregulating autophagy with age, across tissues in *C. elegans*. Specifically, the authors show mir-83/mir-29 to be a secreted cross-tissue regulator of decline in macroautophagy with age. The authors also show mir-83 to be upregulated with age, primarily in the distal intestine, in an HSF-1 dependent manner. They propose that this intestinal increase in mir-83 with age dysregulates autophagy flux by downregulating cup-5 both in the intestine (cell-autonomously) and in BWM that involves mir-83 being secreted in EVs (cell non-autonomously). Furthermore, the authors demonstrate that genetically reducing mir-83 extends lifespan by inducing autophagy and proteostasis in multiple *C. elegans* tissues. Overall, the work is well conducted, technically sound with proper controls, and the proposed mir-83/mir-29-mediated cross-tissue regulation of autophagy is novel and of significance to both the autophagy and aging research fields.

The authors need to address the following concerns, which will provide further supporting evidence for their main conclusions.

1. mir-83 targets genes involved in autophagy

- While the authors show that cup-5/MCOLN is regulated by mir-83 on Day 5, they did not show that this is an age-dependent phenotype per se. Therefore, it is important to monitor the regulation of cup-5/MCOLN by mir-83 (both mRNA and protein levels) with age. The observation of the fluorescence intensity of the translational cup-5 reporter (Fig. 2e) with age would be helpful.
- For the regulation of atg-4.2/ATG4C by mir-83 the authors have not tested atg-4.2/ATG4C levels upon mutating the mir-83 target sequence, as done in Figure 2a for cup-5/MCOLN, which is important for supporting their conclusion that atg-4.2/ATG4C is not an in vivo target of mir-83. Additionally, are mammalian MCOLN and ATG4C also predicted targets of mir-29? A discussion of the divergent regulation of the two predicted autophagy genes by mir-83 is encouraged, especially from an evolutionary standpoint.
- It is unclear whether the dual reporter for monitoring microRNA-3'-UTR interaction is novel, and could be described better.

2. Autophagy measures

- The authors measure autophagy flux in the distal/posterior part of the intestine. It would be important to monitor the autophagy in anterior/proximal intestinal region as control, since mir-83 does not seem to be increased with age in this region (from Images 1c). Higher resolution images of mir-83 expression with age should be included in the supplements (this is also important to convince the reader that mir-83 is not expressed in body-wall muscles; see also Point 5 below regarding new paper by Dzakah et al.).
- The counts in Figure 3c and Figure 3f are very different in absolute numbers. Why?
- Authors must measure autophagy in the neurons as in Figure 3a, b. (A neuronal gfp::lgg-1 strain and tandem lgg-1 strain is available).
- What is the age-point at which sqst-1 mRNA levels were monitored? Importantly, authors need to test if sqst-1 levels are directly regulated by mir-83 (3'-UTR-mediated) or in autophagy-dependent manner only (Figure 3d, 4d).
- Autophagy flux measures are lacking in Figure 3e.
- Importantly, to conclude the cell-autonomous regulation of cup-5 in the intestine by mir-83 (line 185-187), authors need to perform intestine-specific knockdown of cup-5 in Figure 3f.

3. Role of mir-83 in intestine-to-BWM cross-talks

- It is unclear from their images in Fig 2e whether cup-5 is endogenously expressed in BWM. Is CUP-5 upregulated in BWM (Figure 2e) upon mutating mir-83?
- An important control would be to use a muscle-specific RNAi strain and reduce cup-5 in BWM only.

- Additionally, BWM cup-5u reporter levels in the wild-type control vs IntOE of mir-83 is very marginal (Figure 5e), though shown to be significant. This might implicate a secondary signal being involved in intestine-muscle crosstalk.
- Figure 5d should be quantified as a comparison to day 1 as in Figure 5e.

4. Secretion of mir-83

- In Figure 6b, mir-83 levels must be normalized to wild-type level in worms, lower level of mir-83 is therefore expected in BWM compared to whole worms, as mir-83 is not expressed in BWM.
- To conclude that mir-83 is detected in coelomocytes, where it is not expressed, authors need to elaborate on the Ct values observed (what is the standard range for microRNAs?) and how was the statistical analysis done in Figure 6c, f? Also, authors must discuss the design of mir-83 primers used in qPCR. Statistics must be included for Figure S5 b and c. A specific concern here is the mir-83 signal measured in BWM cells from the mir-83(-) null mutant – since the graph is on a logarithmic scale, it is not possible to assess what a signal in this condition really means (see also comment below in Point 5 regarding Dzakah paper).
- Throughout the manuscript, the nature of the mir-83 mutation is not clear. Is it a mutation or a deletion? How is mir-83 still detected in the mir-83 mutant in Figure S5b (red panel). This is a critical piece of information, which was considered a severe lack throughout the entire manuscript.
- The authors should mention the novelty in collecting EVs, if this is a newly developed protocol in *C. elegans*. The authors should include further controls for known cargo EVs as controls. Furthermore, the authors should better elaborate on U18 and whether that is known to be contained in *C. elegans* EVs. Overall, this part of the study is deemed very interesting, but the novelty of it is not clearly communicated.

5. Role of mir-83 in lifespan regulation

- The authors need to discuss a recently published paper by Dzakah et al., 2018 on the role of mir-83 in lifespan regulation. In Dzakah et al., the authors show that downregulating mir-83 extends lifespan (similar to this study). However, Pmir-83::GFP was shown to decrease with age in this study. Additionally, Dzakah et al. suggest that mir-83 is ubiquitously expressed, including in muscle although this is not clearly discerned from the paper. Given that this is a major premise point of the current study, the authors are encouraged to investigate this in more detail, e.g. by isolating the BMW cells from their translational mir-83::mCherry strain and probing for mCherry. Finally, it is shown that hlh-30 is regulated by mir-83. Altogether, it becomes important for Zhou et al. to discuss Dzakah et al. to include how it corroborates or confounds their own findings.
- To meet field standard, the authors are encouraged to tabulate their lifespan analysis as separate experiments, not pooled ones as in Table S1. This is more informative regarding variability between experiments.

6. hsf-1-mir-83-cup-5-hlh-30 axis in regulation of LGG-1

The authors attempt to describe a hsf-1-mir-83-cup-5-hlh-30 regulatory axis. However the data, especially in regard to hlh-30 does not fully support this claim. To address this concern, the authors need to monitor the regulation of cup-5 by hlh-30 as well as the regulation of LGG-1 by mir-83 and hlh-30 with age. Also, how does hsf-1 being upstream of mir-83 modulate LGG-1 levels with age? Authors could monitor the LGG-1 levels in hsf-1;hlh-30 double condition. In addition, it would be important to note whether hsf-1 increases with age throughout the intestine or is limited to the distal intestine. That said, this reviewer does not think that the hlh-30 studies are necessary for the important findings the authors demonstrate in this manuscript and would suggest to remove the hlh-30 data from this manuscript.

7. mir-83 is induced with age

The authors begin this study with the description of their RNA-Seq results, but it remains unclear why the authors chose to further investigate mir-83. Was mir-83 the most differentially upregulated microRNA at day 21? Discussion of the transcriptomics data is lacking. Additionally, the age-points for

qPCR validation of increased mir-83 levels with age (Figure 1a) should include at least one more time point, e.g. day 14 also noting that the transcriptomics data in Figure S1a was done at multiple time points. Likewise, the authors show mir-83 to be regulated by hsf-1 on Day 7 and HSF-1 levels to increase on day 8 compared to day 1, yet this correlation is not tested in any other time-points.

8. Abstract and Introduction section

The authors' premise for this study is to understand the mechanisms by which autophagy fails with age. The failure of autophagy over time is a hypothesis that is supported by several lines of evidence, but it remains a hypothesis. The authors should consider rephrasing and aligning their work accordingly, instead using the opportunity for appropriately emphasizing novelty.

Furthermore, one of the strengths of this work is the inter-tissue communication, which could be better emphasized in the abstract, as well as in the introduction. To emphasize the major gaps in known tissue-specific changes in autophagy with age, the authors are directed to a recent review by Hansen et al. 2018 for the known spatiotemporal analysis of autophagy in aging and disease across species.

In the Introduction, the authors should elaborate more on the regulation of autophagy by microRNAs, especially in the context of lifespan (reviewed in Lapierre et al. 2015). The second paragraph, which details the relevance of proteostasis could be more concise, and concept/question driven.

Minor points

1. In the Introduction, reference for enhanced autophagy in various longevity model is missing.
2. Authors need to clarify the RNA-sequencing method on pg 5, whether it is mRNA-seq or micro-RNA seq?
3. Stats for Figure 1e and Figure 7 is missing in the legend on pg34 and 45.
4. Authors have not commented on the use of luc-2i as control, instead of L4440 vector control used in the field.
5. Is it proximal intestine for lysosome counts in Figures3 b, c? and lysosomal reporters are incorrectly labelled here.
6. Exposure time must be included for the reporter assays.
7. What day was the flux assay conducted in Figure 4c? Missing in text (Pg189) as well as from legend (Pg38).
8. In Figure 5d, authors need to normalize the cup-5u reporter levels to wildtype day1. And, legends for 5c and 5d are interchanged.
9. Demarcate the vacuole area being measured in Figure 6e.
10. Figure S5b need to be better represented using box to separate different genotypes.
11. Revise Table 1 to show independent lifespan data sets.
12. In the method section on Pg23, it is mentioned that BWM cells or coelomocytes were collected in PBS and not Trizol?
13. The predicted targets of mir-83, i.e., cup-5/MCOLN and atg-4.2/ATG4C are abruptly mentioned in the results on pg 5, but should rather be part of Figure 2 and the text accordingly.
14. Line 51: machinery should be replaced by mechanism
15. Figure 2c, e, the genotype should be fully described to make it clear when a transcriptional and when a translational reporter is used.

Reviewer #2 (Remarks to the Author):

This manuscript examines the hypothesis that a miRNA up-regulated in the intestine during aging can directly regulate targets in distant tissues, through extracellular miRNA transport mechanisms. The physiological context is coordinated autophagy with ageing, which is known to occur in the model organism *C.elegans*. The authors provide new knowledge to link coordinated autophagy to the regulation of CUP-5 by miR-83.

Overall the work is interesting and the authors take advantage of the genetic tools available in *C. elegans* to demonstrate convincingly the relevance of miR-83 in regulation of autophagy via targeting of Cup5.

The findings are novel: there are limited examples of functional roles of extracellular miRNAs in model organisms and the results here are rather convincing. There are a few issues that should be clarified or addressed prior to publication, detailed below.

1) Central to this work is the ability to distinguish resident (produced inside the cell) from mobile (imported) miRNA. This is challenging – the authors rely on comparison of miR-83 transcriptional activity (GFP reporter under miR-83 promoter) versus the mature miRNA level by qRT-PCR. Since miRNAs can have very long half-lives presumably even low level transcriptional activity could result in meaningful miRNA abundance. Further explanation/discussion of caveats is warranted. It would of course be hugely beneficial to identify genes that block miRNA transport as a way of validating results (I appreciate this may be beyond the scope of this work).

2) Related to comment 1, the authors do provide additional evidence for miR-83 transfer from the intestine to the body wall based on detection of miR-83 in body wall when using an intestine-specific transgene expressing miR-83 (in context of miR-83 (-) animals). While this is compelling, it would be helpful to provide more background on the technology and assurance this transgene would not express at a low level in the body wall. Perhaps a complementary approach would be to inject miR-83 straight into the intestine and see if this regulates Cup5 3'UTR reporter in the body wall.

3) A key issue in extracellular miRNA field is specificity: are certain miRNAs selected for export over others (implying degrees of control). It would be useful to understand whether the transport of any miRNA from the intestine to the BWM can occur based on intestine-specific overexpression, or whether this is specific to miR-83.

Minor:

4) Whilst this work is a nice proof of concept of mobile miRNA function, with convincing physiological relevance, it is targeted to proving one particular target that may have been selected out of hundreds. This is a very biased way to represent the function of the secreted miRNA, which might influence other aspects of nematode physiology. This bias should be acknowledged in the discussion, or a better demonstration that autophagy is the only pathway regulated by secreted miR-83 should be provided.

5) A better explanation/diagram of the role of coelomocytes in transfer of miRNA from the intestine to the BWM would be helpful for non-worm experts (these are not included in model in Figure 7i).

6) The “relative expression” of miR-83 in figures should be qualified – what is this normalized to? – eg. Figure 1a.

Reviewer #3 (Remarks to the Author):

Shen and colleagues identify a miRNA-gene regulatory circuit that acts cell non-autonomously to regulate autophagy, protein homeostasis and aging in *C. elegans*. These exciting results will surely be impactful and will appeal to a broad audience.

Major Points

A little more rigor is needed to confirm that miR-83 is transported across tissues. That GFP expressed under the mir-83 promoter is not visible in the body wall muscle, particularly at the low magnification shown in the figures, is not convincing proof that miR-83 is not expressed in body wall muscle cells. qPCR of the endogenous primary transcript and precursor of miR-83 should be done using the same single cell sequencing approaches used in Figure 6, including the isolated body wall muscle cells, coelomocytes, and extracellular vesicles. As the authors presumably already have this RNA in hand, it does not seem too burdensome. This would also help to clarify whether it's the mature miR-83 or pre-miR-83 that is mobile.

Minor Points

Statistical Analyses:

1. All bars in the plots should have error bars. It's unclear why the wild type or control bars in many of the plots do not have them.
2. I think that standard deviation should be used in place of the standard mean. The samples are pools of numerous individuals and thus each sample already represents a population mean. Thus, standard deviation should be used to report variability in sampling.
3. As western blot signals were quantified in Photoshop, it's important that the authors confirm that the chemiluminescence signals were not saturated, as they appear to be in some of the blot images.
4. The ANOVA test assumes normal distribution and equal variance of data. This is often not the case with count data and thus I suspect that the test is inappropriately used in many of the analyses (e.g. Figures 3 and 4). The authors should confirm the ANOVA assumptions are met, or use a different test.

Other Concerns:

4. Figure S2a cited out of order.
5. Figure S1. What do the p values in Figure S1a correspond to - comparison to day 1?
6. Figure S1. What RNA-seq datasets are analyzed here? The original papers should be cited. Data analysis should also be described.
7. Figure 1 and S1. The qPCR results in Figure 1a don't seem consistent with the small RNA sequencing results in Figure S1a. By qPCR, there appears to be a significant change in miR-83 levels but by RNA-seq there is no change out to day 14. This is troubling because the authors claim to have honed in on miR-83 based on the RNA-seq data.
8. Figure 1. The western blots in Figure 1b are a bit concerning because the GFP blot is truncated and the α -tubulin blot appears saturated.
9. Figure 1. The GFP images in Figure 1c suggest a much greater than 2-fold increase in GFP expression between day 7 and day 14.
10. Figure 1. The hsf-1 RNAi treated animals IN Figure 1f appear stunted in their development. The authors should note that development arrest or delay could indirectly affect miR-83 expression.
11. Figure 4. It is not clear why over expressing miR-83 causes GFP-LGG-1 foci to increase to the

same level as what is observed in the mir-83 loss of function mutant in Figure 4e. Isn't this the opposite of what would be predicted? Same issue as in 4e.

12. Figure S5. It's not clear why miR-83 is detected in the mir-83 mutant line in Figure S5b.

13. Figure S5a cited out of order.

14. Without reading the methods section, it is unclear what is being measured in the miR-83 qPCR assays in Figures 6 and S5. In the results section and in the figures, when referring to the mature miR-83, the syntax miR-83 should be used. When referring to the mir-83 gene, lower case italics should be used.

15. Lines 239-240. "...and the contamination from the two mir-83 enriched tissues (i.e., neurons and 240 intestine) was excluded." It's not clear what this means.

16. Methods: The specific antibodies, not just the manufacturers, should be noted.

Our detailed responses to the reviewers' concerns are listed below on a point-by-point basis.

Reviewer #1 (Remarks to the Author):

In this interesting manuscript, Zhou et al. study the role of microRNAs in dysregulating autophagy with age, across tissues in *C. elegans*. Specifically, the authors show mir-83/mir-29 to be a secreted cross-tissue regulator of decline in macroautophagy with age. The authors also show mir-83 to be upregulated with age, primarily in the distal intestine, in an HSF-1 dependent manner. They propose that this intestinal increase in mir-83 with age dysregulates autophagy flux by downregulating cup-5 both in the intestine (cell-autonomously) and in BWM that involves mir-83 being secreted in EVs (cell non-autonomously). Furthermore, the authors demonstrate that genetically reducing mir-83 extends lifespan by inducing autophagy and proteostasis in multiple *C. elegans* tissues. Overall, the work is well conducted, technically sound with proper controls, and the proposed mir-83/mir-29-mediated cross-tissue regulation of autophagy is novel and of significance to both the autophagy and aging research fields.

The authors need to address the following concerns, which will provide further supporting evidence for their main conclusions.

1. mir-83 targets genes involved in autophagy

- While the authors show that cup-5/MCOLN is regulated by mir-83 on Day 5, they did not show that this is an age-dependent phenotype per se. Therefore, it is important to monitor the regulation of cup-5/MCOLN by mir-83 (both mRNA and protein levels) with age. The observation of the fluorescence intensity of the translational cup-5 reporter (Fig. 2e) with age would be helpful.

-- Many thanks for the comment! From confocal microscopy on the translational *cup-*

5 reporter, CUP-5 is regulated by *mir-83* in the intestine and BWM of both young and aged worms (Fig. 2d). We further knocked in a 3xFLAG::WrmScarlet tag at the N-terminal of *cup-5* by Crispr/Cas9 technology for a more precise quantification. Using this endogenously labelled *cup-5*, we examined the age-related regulation of CUP-5 by *mir-83* by Western blot. The endogenously labelled CUP-5 was suppressed by *mir-83* in both young and old worms. Consistent with the upregulation of *mir-83* with ageing, we observed a stronger regulation in day 7 adult worms (Fig. 2e). Since microRNAs do not inhibit gene expression only by degrading target mRNA, we did not examine the mRNA level of *cup-5* in the original manuscript. The mRNA level of *cup-5* has now been examined following your comments. It is unchanged with age, with a mild but insignificant increase upon *mir-83* mutation (Supplementary Fig. 2e), implying that *mir-83* inhibits *cup-5* mainly by blocking its translation.

- For the regulation of *atg-4.2/ATG4C* by *mir-83* the authors have not tested *atg-4.2/ATG4C* levels upon mutating the *mir-83* target sequence, as done in Figure 2a for *cup-5/MCOLN*, which is important for supporting their conclusion that *atg-4.2/ATG4C* is not an in vivo target of *mir-83*.

-- Corresponding experiments have been performed as suggested. The 3'-UTR reporter of *atg-4.2* with mutated *mir-83* target sequence does not respond to *mir-83* overexpression in HEK293T cells (Supplementary Fig. 2b).

Additionally, are mammalian MCOLN and ATG4C also predicted targets of *mir-29*? A discussion of the divergent regulation of the two predicted autophagy genes by *mir-83* is encouraged, especially from an evolutionary standpoint.

-- miR-29 is predicted to target mammalian ATG4D (the mammal ortholog of *atg-4.2*), but not MCOLN. In addition, it is also predicted to bind to the 3'-UTRs of

ATG9A and TFEB (Fig. 1 for reviewer).

Figure 1 for reviewer. Zhou et al.

Therefore, we think that the mammalian homolog of *mir-83* also functions in autophagy, although by targeting other critical components. Corresponding discussions have been included in the revised manuscript.

- It is unclear whether the dual reporter for monitoring microRNA-3'-UTR interaction is novel, and could be described better.

-- The reporter is not novel. It is designed following the principle reported in our previous publications^{1,2}. A more detailed description is now included.

2. Autophagy measures

- The authors measure autophagy flux in the distal/posterior part of the intestine. It would be important to monitor the autophagy in anterior/proximal intestinal region as control, since *mir-83* does not seem to be increased with age in this region (from Images 1c).

-- From images of higher resolution (Supplementary Fig. 1c-d), *mir-83* is also expressed and increased in the anterior/proximal intestinal region with age. Therefore, this region cannot serve as a control as suggested.

Higher resolution images of *mir-83* expression with age should be included in the supplements (this is also important to convince the reader that *mir-83* is not expressed in body-wall muscles; see also Point 5 below regarding new paper by Dzakah et al.).

-- We examined *mir-83* expression with age at higher resolution (Supplementary Fig. 1c-d) as suggested. We did not see any expression of *mir-83* in the body wall muscle. Please see our response to “Point 5” for more details.

- The counts in Figure 3c and Figure 3F are very different in absolute numbers. Why?

-- Worms were fed with OP50 in Fig. 3c and HT115 in Fig. 3f (Fig. 3d in the revised manuscript). The different nutrients in these two bacteria strains were reported to have metabolic impact on worms and may influence autophagy and contribute to the different numbers of APs and ALs³.

- Authors must measure autophagy in the neurons as in Figure 3a, b. (A neuronal *gfp::lgg-1* strain and tandem *lgg-1* strain is available).

-- Autophagy in the neurons were measured as suggested, using a neuron specific reporter of *mCherry::GFP::LGG-1* (Fig. 3g and h). Consistent with our previous results, autophagy in neurons is not significantly affected in *mir-83(-)* mutants.

- What is the age-point at which *sqst-1* mRNA levels were monitored? Importantly, authors need to test if *sqst-1* levels are directly regulated by *mir-83* (3'-UTR-mediated) or in autophagy-dependent manner only (Figure 3d, 4d).

-- Thanks for the suggestion! We measured *sqst-1* mRNA levels at day 1, day 7, and day 14 of adulthood (Supplementary Fig. 3b). It is unchanged by either ageing or *mir-83* mutation. In addition, the 3'-UTR of *sqst-1* does not contain *mir-83* binding site by TargetScan prediction⁴.

- Autophagy flux measures are lacking in Figure 3e.

-- The autophagy flux was measured as suggested (Fig. 3f).

- Importantly, to conclude the cell-autonomous regulation of *cup-5* in the intestine by *mir-83* (line 185-187), authors need to perform intestine-specific knockdown of *cup-5* in Figure 3f.

-- Thanks for your suggestion! The intestine-specific RNAi of *cup-5* was performed and indeed blocked the enhanced autophagy in the intestine of *mir-83(-)* mutants, indicating a cell-autonomous regulation of *cup-5* in the intestine by *mir-83* (Fig. 3d).

3. Role of *mir-83* in intestine-to-BWM cross-talks

- It is unclear from their images in Fig 2e whether *cup-5* is endogenously expressed in BWM. Is CUP-5 upregulated in BWM (Figure 2e) upon mutating *mir-83*?

-- *cup-5* is reported to be endogenously expressed in BWM⁵. Using the translational reporter of *cup-5*, we also examined CUP-5 expression with confocal microscopy. As shown in Fig. 2d, CUP-5 is indeed expressed in BWM and upregulated upon mutating *mir-83*.

- An important control would be to use a muscle-specific RNAi strain and reduce *cup-5* in BWM only.

-- Thanks for your suggestion! The BWM-specific RNAi of *cup-5* was performed as suggested (Fig. 5a). Inhibiting *cup-5* specifically in BWM completely blocked the enhanced autophagy in *mir-83(-)* mutants, further confirming that *mir-83* controls BWM autophagy via inhibiting CUP-5 in this tissue.

- Additionally, BWM *cup-5u* reporter levels in the wild-type control vs IntOE of *mir-83* is very marginal (Figure 5e), though shown to be significant. This might implicate a secondary signal being involved in intestine-muscle crosstalk.

-- We agree with your speculation. In our eyes, this “secondary signal” could be from the machinery transporting *mir-83* from the intestine into BWM. For example, a negative feedback within (e.g., the scavenging of *mir-83* from the pseudocoelom fluidics by coelomocytes) could reduce the effect of *mir-83* IntOE so that the overexpression of *mir-83* in the intestine of WT worms only inhibits the *cup-5u* reporter in BWM mildly but significantly. Corresponding discussion has been included in the manuscript.

- Figure 5d should be quantified as a comparison to day 1 as in Figure 5e.

-- Quantification has been performed as suggested (Fig. 5d). The original Fig. 5d and Fig. 5e are from the same experiment but split into two panels. Since the quantification is now all normalized against day 1 WT samples, the two panels are now merged and presented with the unsplit blots.

4. Secretion of *mir-83*

- In Figure 6b, mir-83 levels must be normalized to wild-type level in worms, lower level of mir-83 is therefore expected in BWM compared to whole worms, as mir-83 is not expressed in BWM.

-- The cDNAs for single cell qPCR and worm qPCR were prepared following different protocols as shown in the “Methods”. Simply speaking, cDNA was subjected to an additional step of PCR amplification for single cell qPCR (e.g. BWM cells). Therefore, we do not think it proper to normalize the *mir-83* level in BWM cells to that in worms. By qPCR, it is difficult to compare the absolute amounts of *mir-83* in BWM and worms.

- To conclude that mir-83 is detected in coelomocytes, where it is not expressed, authors need to elaborate on the Ct values observed (what is the standard range for microRNAs?) and how was the statistical analysis done in Figure 6c, f?

-- We think Ct values below 40 indicate that a gene is detected by qRT-PCR. To give the readers more ideas about the Ct values of microRNAs in coelomocytes, we examined *mir-56* and *mir-82*, two microRNAs abundant in extracellular vesicles but not transcribed in coelomocytes^{6,7}, in the isolated coelomocytes (Fig. 6c). They were also detected with Ct values of 25.91 and 25.67 respectively.

As for the statistical analysis in Fig. 6c and f, we did not perform any analysis. In these assays, we do not intend to compare the different levels of detected signals but to detect the presence of a certain gene.

Also, authors must discuss the design of mir-83 primers used in qPCR. Statistics must be included for Figure S5 b and c.

-- The design of *mir-83* primers was as reported⁸. Statistics have been included for

Supplementary Fig. 5b and c. Please see Supplementary Table 2.

A specific concern here is the *mir-83* signal measured in BWM cells from the *mir-83(-)* null mutant – since the graph is on a logarithmic scale, it is not possible to assess what a signal in this condition really means (see also comment below in Point 5 regarding Dzakah paper).

-- We are sorry for the confusion! For those sample without qPCR signal (e.g. *mir-83* in *mir-83(-)* mutants), their Ct values were arbitrarily set as 40 because the qPCR reactions ended at cycle 40. This then resulted in the confusing data presentation.

In the original Supplementary Fig. 5b, the relative transcription levels were calculated by the following formula:

$$0.5^{[(Ct_{\text{mir-83}} - Ct_{\text{U18}})_{\text{BWM}} - (Ct_{\text{mir-83}} - Ct_{\text{U18}})_{\text{worm}}]}$$

In the revised Fig. 5b, instead of the confusing “relative transcription levels”, both the mean and individual Ct values from single cell qPCR experiments were shown. As you can see, no *mir-83* was detected (Ct = 40(ND)) in the BWM or worm of *mir-83(-)* mutants.

- Throughout the manuscript, the nature of the *mir-83* mutation is not clear. Is it a mutation or a deletion?

-- It is a null mutation, deleting the entire *mir-83* gene. Corresponding information is included in the revised manuscript.

How is *mir-83* still detected in the *mir-83* mutant in Figure S5b (red panel). This is a critical piece of information, which was considered a severe lack throughout the

entire manuscript.

-- We are again sorry for our confusing data presentation in the original Supplementary Fig. 5b. As just mentioned above, Ct values of samples without qPCR signals were arbitrarily set as 40 to facilitate the original quantification of “relative transcription levels”. In fact, *mir-83* was not detected in *mir-83(-)* mutants in all experiments (Fig. 5b). The data are now presented with the exact Ct values to avoid misunderstanding. Samples without qPCR signals are now labeled as 40(ND).

- The authors should mention the novelty in collecting EVs, if this is a newly developed protocol in *C. elegans*.

-- Many thanks for appreciating this protocol! The idea underlying this method is that a proportion of EVs are secreted out of the worm as reported⁹⁻¹¹. Following this idea, EVs were collected from the plate in a previous report⁹. But this method also collects EVs from bacteria. To minimize this contamination, we designed this protocol by soaking washed worms in M9 and collected EVs from M9. Corresponding information is now included in the modified ‘Methods’ section.

The authors should include further controls for known cargo EVs as controls. Furthermore, the authors should better elaborate on U18 and whether that is known to be contained in *C. elegans* EVs. Overall, this part of the study is deemed very interesting, but the novelty of it is not clearly communicated.

-- We are very glad to learn your appreciation of this part of our study. Following your suggestion, we examined the levels of *mir-56* and *mir-82*, two microRNAs reported to be highly abundant in EVs⁷, as controls. As shown in Fig. 6f, we also detected a high level of these two microRNAs in our EV samples, confirming the quality of our EV preparation.

U18 is a snoRNA commonly used as the reference in small RNA qRT-PCR of worm samples. We found that this snoRNA is indeed abundant in EVs, consistent with the finding that the snoRNA U6 is highly expressed in mammalian exosomes¹². But since little is known about the components of *C.elegans* EVs, there is no widely accepted reference genes in qRT-PCR assays on worm EV samples. Therefore, we present the Ct values in the revised manuscript instead of the confusing relative quantification in the previous version.

5. Role of mir-83 in lifespan regulation

- The authors need to discuss a recently published paper by Dzakah et al., 2018 on the role of mir-83 in lifespan regulation. In Dzakah et al., the authors show that downregulating mir-83 extends lifespan (similar to this study). However, Pmir-83::GFP was shown to decrease with age in this study.

-- As suggested, we examined the expression of *mir-83* at more time points and found it still upregulated with age, using multiple methods (RNA-Seq, qPCR, and transcriptional reporters) (Fig. 1 and Supplementary Fig. 1). Our result is also consistent with previous RNA-Seq studies by Kato et al¹³.

In the recent report by Dzakah et al., FUdR was supplemented in the NGM plates for collecting worms at different ages¹⁴. To collect worms for qPCR, Dzakah et al. transferred Day 1 adult worms onto FUdR plates. Therefore, when analysing *mir-83* expression with ageing, Dzakah et al. compared aged worms undergoing days of FUdR treatment to Day 1 young adults with a very brief or without FUdR treatment. Because no detailed information was provided in their manuscript, we suppose their assays with a *mir-83* transcription reporter were performed in a similar manner.

FUdR is known to have many effects in ageing (e.g. modulating proteostasis and mitochondria)¹⁵⁻¹⁷. The comparison between sustained FUdR-treated worms and

worms undergoing brief or no FUdR treatment is also inappropriate in our eyes. Therefore, we did not use any FUdR in our experiments. We think the decreased *mir-83* expression with ageing observed by Dzakah et al. may be due to some unknown effects by FUdR.

Additionally, Dzakah et al. suggest that *mir-83* is ubiquitously expressed, including in muscle although this is not clearly discerned from the paper. Given that this is a major premise point of the current study, the authors are encouraged to investigate this in more detail, e.g. by isolating the BWM cells from their translational *mir-83::mCherry* strain and probing for mCherry.

-- We agree that this is a key point of our study but would also like to clarify that our exact argument is that *mir-83* is a secreted microRNA and controls autophagy in BWM non-autonomously.

As for the discrepancy with the report from Dzakah et al, first, as you also mentioned, we are not able to see clear *mir-83* expression in BWM in their published data, either.

Second, our results using both a published transgene with 1,978 bp of *mir-83* promoter⁶ or a homemade transgene with 3,110 bp of *mir-83* promoter do not show any *mir-83* expression in BWM (Fig. 1 and Supplementary Fig. 1), which is consistent with previous reports from Walhout lab and Ambros lab^{6,18}.

Third, we performed the experiments as you suggested. As shown in Supplementary Fig. 5, *gfp* from *myo-3p::gfp* but not *TagRFP* from *mir-83p::TagRFP* was detected by qPCR in the isolated BWM cells, further confirming that *mir-83* is not transcribed in the BWM.

Taken together, our data support that *mir-83* is not expressed in BWM, which is consistent with multiple previous reports^{6,18}. Nevertheless, we cannot exclude that a trace amount of *mir-83* below the detection limit in our assays is transcribed in BWM

as commented by Reviewer #2 and implied by Dzakah et al.¹⁴. We therefore examined the effect of an intestine-specific transgene of *mir-83* in *mir-83(-)* mutants. As shown in Fig. 6 and S5, *mir-83* but not *gfp* driven by the intestine-specific promoter is detected in isolated BWM cells. Together with its detection in EVs and other phenotypes by the intestine-specific transgene of *mir-83*, we therefore conclude that *mir-83* is a secreted microRNA and controls autophagy non-autonomously in BWM. Corresponding results and discussion have been included in the revised manuscript.

Finally, it is shown that *hlh-30* is regulated by *mir-83*. Altogether, it becomes important for Zhou et al. to discuss Dzakah et al. to include how it corroborates or confounds their own findings.

-- *hlh-30* does not harbor *mir-83* binding sites in its 3'-UTR. Dzakah et al. observed an upregulation of *hlh-30* mRNA in *mir-83* mutants but did not show any more evidence with HLH-30 protein level or a 3'UTR reporter of *hlh-30*. Therefore, their report does not support *hlh-30* as a direct target gene of *mir-83*. The increase of *hlh-30* transcription should be an indirect effect by *mir-83*. Nevertheless, this piece of data by Dzakah et al. is consistent with our finding that *hlh-30* may function downstream of *mir-83* in the *mir-83-cup-5-hlh-30* axis (now removed from the manuscript as you suggested).

Discussion of the report by Dzakah et al. is included in the revised manuscript.

- To meet field standard, the authors are encouraged to tabulate their lifespan analysis as separate experiments, not pooled ones as in Supplementary Table 1. This is more informative regarding variability between experiments.

-- Lifespan analysis is now shown as suggested in the modified Supplementary Table

1.

6. hsf-1-mir-83-cup-5-hlh-30 axis in regulation of LGG-1

The authors attempt to describe a hsf-1-mir-83-cup-5-hlh-30 regulatory axis. However the data, especially in regard to hlh-30 does not fully support this claim. To address this concern, the authors need to monitor the regulation of cup-5 by hlh-30 as well as the regulation of LGG-1 by mir-83 and hlh-30 with age. Also, how does hsf-1 being upstream of mir-83 modulate LGG-1 levels with age? Authors could monitor the LGG-1 levels in hsf-1;hlh-30 double condition. In addition, it would be important to note whether hsf-1 increases with age throughout the intestine or is limited to the distal intestine. That said, this reviewer does not think that the hlh-30 studies are necessary for the important findings the authors demonstrate in this manuscript and would suggest to remove the hlh-30 data from this manuscript.

-- We agree with your comments. *hlh-30* data are removed from the manuscript as suggested.

7. mir-83 is induced with age

The authors begin this study with the description of their RNA-Seq results, but it remains unclear why the authors chose to further investigate mir-83. Was mir-83 the most differentially upregulated microRNA at day 21? Discussion of the transcriptomics data is lacking.

-- *mir-83* is one of the five conserved microRNAs upregulated with ageing in our RNA-Seq results, although not the most differentially upregulated ones. In addition, this family also target important genes in autophagy. That is why *mir-83* was picked. We mentioned our RNA-Seq analysis to give the readers a clue how the study started.

But because this manuscript focuses only on *mir-83*, we think it not necessary to discuss the details of our RNA-Seq data.

Additionally, the age-points for qPCR validation of increased *mir-83* levels with age (Figure 1a) should include at least one more time point, e.g. day 14 also noting that the transcriptomics data in Figure S1a was done at multiple time points.

-- Thanks for your suggestion! qPCR of *mir-83* was performed at more time points as suggested (Fig. 1a).

Likewise, the authors show *mir-83* to be regulated by *hsf-1* on Day 7 and HSF-1 levels to increase on day 8 compared to day 1, yet this correlation is not tested in any other time-points.

-- Thanks for the suggestion! More time points were tested as suggested. The protein level of HSF-1 is increased from day 4 of adulthood (Supplementary Fig. 1f). Consistently, RNAi against *hsf-1* abolishes the increased *mir-83* expression at Day 4 or Day 7 (Fig. 1e-f).

8. Abstract and Introduction section

The authors' premise for this study is to understand the mechanisms by which autophagy fails with age. The failure of autophagy over time is a hypothesis that is supported by several lines of evidence, but it remains a hypothesis. The authors should consider rephrasing and aligning their work accordingly, instead using the opportunity for appropriately emphasizing novelty.

Furthermore, one of the strengths of this work is the inter-tissue communication, which could be better emphasized in the abstract, as well as in the introduction. To

emphasize the major gaps in known tissue-specific changes in autophagy with age, the authors are directed to a recent review by Hansen et al. 2018 for the known spatiotemporal analysis of autophagy in aging and disease across species.

In the Introduction, the authors should elaborate more on the regulation of autophagy by microRNAs, especially in the context of lifespan (reviewed in Lapierre et al. 2015). The second paragraph, which details the relevance of proteostasis could be more concise, and concept/question driven.

-- Many thanks for the suggestions! The manuscript has been modified accordingly.

Minor points

1. In the Introduction, reference for enhanced autophagy in various longevity model is missing.

-- The corresponding reference has been added.

2. Authors need to clarify the RNA-sequencing method on pg 5, whether it is mRNA-seq or micro-RNA seq?

-- It is a microRNA sequencing. The information is clarified in the revised manuscript.

3. Stats for Figure 1e and Figure 7 is missing in the legend on pg34 and 45.

-- Sorry for the mistakes. Corresponding information is now included in the figure legends.

4. Authors have not commented on the use of luc-2i as control, instead of L4440 vector control used in the field.

-- We think RNAi against a real gene which is not expressed in worm would provide a better RNAi control. Corresponding comments are included in the modified 'Methods' section.

5. Is it proximal intestine for lysosome counts in FigureS3 b, c? and lysosomal reporters are incorrectly labelled here.

-- Yes. The information is included in the revised legend. The figure is also modified as suggested.

6. Exposure time must be included for the reporter assays.

-- Corresponding information is included in the "Source Data".

7. What day was the flux assay conducted in Figure 4c? Missing in text (Pg189) as well as from legend (Pg38).

-- It was performed at day 1 as shown in the figure. Corresponding information is included in the revised legend and text.

8. In Figure 5d, authors need to normalize the cup-5u reporter levels to wildtype day1. And, legends for 5c and 5d are interchanged.

-- The manuscript has been modified as suggested.

9. Demarcate the vacuole area being measured in Figure 6e.

-- Fig. 6e has been noted as suggested.

10. Figure S5b need to be better represented using box to separate different genotypes.

-- The figure is modified as suggested.

11. Revise Table 1 to show independent lifespan data sets.

-- The table is modified as suggested.

12. In the method section on Pg23, it is mentioned that BWM cells or coelomocytes were collected in PBS and not Trizol?

-- The isolated cells were first collected in PBS and then subjected to lysis by adding lysis buffer. Please also see the 'Quantification RT-PCR' in 'Methods'.

13. The predicted targets of mir-83, i.e., cup-5/MCOLN and atg-4.2/ATG4C are abruptly mentioned in the results on pg 5, but should rather be part of Figure 2 and the text accordingly.

-- The manuscript was modified as suggested.

14. Line 51: machinery should be replaced by mechanism

-- The manuscript was modified as suggested.

15. Figure 2c, e, the genotype should be fully described to make it clear when a transcriptional and when a translational reporter is used.

-- The manuscript was modified as suggested.

Reviewer #2 (Remarks to the Author):

This manuscript examines the hypothesis that a miRNA up-regulated in the intestine during aging can directly regulate targets in distant tissues, through extracellular miRNA transport mechanisms. The physiological context is coordinated autophagy with ageing, which is known to occur in the model organism *C.elegans*. The authors provide new knowledge to link coordinated autophagy to the regulation of CUP-5 by miR-83.

Overall the work is interesting and the authors take advantage of the genetic tools available in *C.elegans* to demonstrate convincingly the relevance of miR-83 in regulation of autophagy via targeting of Cup5.

The findings are novel: there are limited examples of functional roles of extracellular miRNAs in model organisms and the results here are rather convincing. There are a few issues that should be clarified or addressed prior to publication, detailed below.

1) Central to this work is the ability to distinguish resident (produced inside the cell)

from mobile (imported) miRNA. This is challenging – the authors rely on comparison of miR-83 transcriptional activity (GFP reporter under miR-83 promoter) versus the mature miRNA level by qRT-PCR. Since miRNAs can have very long half-lives presumably even low-level transcriptional activity could result in meaningful miRNA abundance. Further explanation/discussion of caveats is warranted.

-- To examine whether low-level transcription of *mir-83* occurs in the BWM cells, we isolate BWM cells from the worms expressing *myo-3p::gfp* and *mir-83p::TagRFP*, following the suggestion from Reviewer 1#. By qPCR, *gfp* from *myo-3p::gfp* but not *TagRFP* from *mir-83p::TagRFP* was detected by qPCR in the isolated BWM cells (Supplementary Fig. 5c-d).

Nevertheless, we agree that a trace amount of *mir-83* could be transcribed in BWM, which is below the detection sensitivity of our current methods. That is why we expressed *mir-83* specifically in the intestine of *mir-83(-)* mutant worms (Fig. 5 and 7), as you also appreciated. To validate the specificity of the *ges-1p::mir-83* transgene, BWM cells were also isolated from worms expressing *myo-3p::gfp* and *ges-1p::TagRFP*. By qPCR, we detected only *gfp* but not *TagRFP* in these cells (Supplementary Fig. 5c-d). Therefore, the phenotypes caused by our intestine-specific transgene of *mir-83* (*ges-1p::mir-83*) (Fig. 5, 6, and 7; Supplementary Fig. 5 and 7) are not due to any expression leakage in the BWM cells, but by the transportation of *mir-83* from the intestine.

It would of course be hugely beneficial to identify genes that block miRNA transport as a way of validating results (I appreciate this may be beyond the scope of this work).

-- We agree that the machinery transporting miRNAs across tissues is a very intriguing and important issue to explore. We also agree that this is beyond the scope of this manuscript. So, we added a short discussion about this issue in the revised

manuscript to incite future studies.

2) Related to comment 1, the authors do provide additional evidence for miR-83 transfer from the intestine to the body wall based on detection of miR-83 in body wall when using an intestine-specific transgene expressing miR-83 (in context of miR-83 (-) animals). While this is compelling, it would be helpful to provide more background on the technology and assurance this transgene would not express at a low level in the body wall. Perhaps a complementary approach would be to inject miR-83 straight into the intestine and see if this regulates Cup5 3'-UTR reporter in the body wall.

-- qPCR of *TagRFP* and *gfp* was performed in the isolated BWM cells from worms expressing *ges-1p::TagRFP* and *myo-3p::gfp*. Only *gfp* was detected in these cells (Supplementary Fig. 5c-d), indicating that the transgene is not expressed in BWM.

3) A key issue in extracellular miRNA field is specificity: are certain miRNAs selected for export over others (implying degrees of control). It would be useful to understand whether the transport of any miRNA from the intestine to the BWM can occur based on intestine-specific overexpression, or whether this is specific to miR-83.

-- We agree with your comments. We detected other microRNAs in coelomocytes and EVs secreted by worms (Fig. 6c and f)^{6,7}, suggesting that the transport of microRNAs across tissues may be common in *C.elegans*. But we think this issue is beyond the scope of this manuscript which focused on *mir-83* only.

Corresponding discussion on the specificity of extracellular microRNAs has been included in the revised manuscript.

Minor:

4) Whilst this work is a nice proof of concept of mobile miRNA function, with convincing physiological relevance, it is targeted to proving one particular target that may have been selected out of hundreds. This is a very biased way to represent the function of the secreted miRNA, which might influence other aspects of nematode physiology. This bias should be acknowledged in the discussion, or a better demonstration that autophagy is the only pathway regulated by secreted miR-83 should be provided.

-- Thanks for your critics! The 'Discussion' section has been modified accordingly.

5) A better explanation/diagram of the role of coelomocytes in transfer of miRNA from the intestine to the BWM would be helpful for non-worm experts (these are not included in model in Figure 7i).

-- We examined *mir-83* in isolated coelomocytes because these cells could enrich molecules in the body cavity fluid. Therefore, the presence of *mir-83* in coelomocytes implies that it is transported from the intestine to BWM through the body cavity fluid. However, we have no idea of the specific role of the coelomocytes in this transfer. We agree that whether these cells regulate *mir-83* transportation by scavenging *mir-83* in the pseudocoelom fluidics or have other functions is very interesting for future study. Corresponding discussion is included in the revised manuscript. But we also think that this is out of the focus of this manuscript. That is why these cells are not included in the model figure.

6) The "relative expression" of miR-83 in figures should be qualified – what is this normalized to? – eg. Figure 1a.

-- Sorry for overlooking this issue. Corresponding information is included in the revised figure legends.

Reviewer #3 (Remarks to the Author):

Shen and colleagues identify a miRNA-gene regulatory circuit that acts cell non-autonomously to regulate autophagy, protein homeostasis and aging in *C. elegans*. These exciting results will surely be impactful and will appeal to a broad audience.

Major Points

A little more rigor is needed to confirm that miR-83 is transported across tissues. That GFP expressed under the *mir-83* promoter is not visible in the body wall muscle, particularly at the low magnification shown in the figures, is not convincing proof that miR-83 is not expressed in body wall muscle cells.

-- We do appreciate your critics. Following the suggestion by Reviewer #1, we used qPCR to examine the expression of *TagRFP* from *mir-83p::TagRFP* and *ges-1p::TagRFP* in isolated BWM cells. As shown in Supplementary Fig. 5, *TagRFP* is not detected in the BWM cells. These results first confirmed that *mir-83* is not expressed in BWM. Second, because the intestine-specific expression of *mir-83* (*ges-1p::mir-83*) rescues the *mir-83(-)* phenotypes in BWM (Fig. 5, 6, and 7; Supplementary Fig. 5 and S7), these results further indicate that this is not due to the expression of *ges-1p::mir-83* in BWM and *mir-83* is transported across tissues.

qPCR of the endogenous primary transcript and precursor of miR-83 should be done

using the same single cell sequencing approaches used in Figure 6, including the isolated body wall muscle cells, coelomocytes, and extracellular vesicles. As the authors presumably already have this RNA in hand, it does not seem too burdensome. This would also help to clarify whether it's the mature miR-83 or pre-miR-83 that is mobile.

-- Many thanks for your suggestions! Corresponding qPCRs have been performed (Supplementary Fig. 5). By these data, the mature miR-83 is the dominant form of *mir-83* transported across tissues. This may be due to the dominant abundance of mature miR-83 in the worm.

Minor Points

Statistical Analyses:

1. All bars in the plots should have error bars. It's unclear why the wild type or control bars in many of the plots do not have them.

-- Thanks for your critics. In some experiments in the original manuscript, all data were normalized twice (first to the reference gene, and second to the control sample) in each replicate. As a result, the mean was "1" and the SD was "0" of control sample when the means of three or more replicates were plotted together. We have modified these data as you commented to improve the statistics in this study.

2. I think that standard deviation should be used in place of the standard mean. The samples are pools of numerous individuals and thus each sample already represents a population mean. Thus, standard deviation should be used to report variability in sampling.

-- The variability of all the results except for those of autophagy markers is now

presented in standard deviation as suggested.

3. As western blot signals were quantified in Photoshop, it's important that the authors confirm that the chemiluminescence signals were not saturated, as they appear to be in some of the blot images.

-- The raw data of western blot signals are 8-bit images captured by a Tanon 5200 Luminescent Imaging Workstation. Only images without saturated bands were subjected to quantification. Corresponding information were included in the 'Western blotting' of the 'Methods' section.

4. The ANOVA test assumes normal distribution and equal variance of data. This is often not the case with count data and thus I suspect that the test is inappropriately used in many of the analyses (e.g Figures 3 and 4). The authors should confirm the ANOVA assumptions are met, or use a different test.

-- Thanks for your critics. Poisson regression is a more suitable model for statistics of puncta counting compared to ANOVA¹⁹. We have corrected these data accordingly.

Other Concerns:

4. Figure S2a cited out of order.

-- This issue has been corrected.

5. Figure S1. What do the p values in Figure S1a correspond to - comparison to day 1?

-- Yes, the two *p* values are versus day 1. Corresponding information is included in the

revised figure legend.

6. Figure S1. What RNA-seq datasets are analyzed here? The original papers should be cited. Data analysis should also be described.

-- It is an RNA-Seq performed by us, not from reported datasets. Corresponding information is included in the revised manuscript.

7. Figure 1 and S1. The qPCR results in Figure 1a don't seem consistent with the small RNA sequencing results in Figure S1a. By qPCR, there appears to be a significant change in miR-83 levels but by RNA-seq there is no change out to day 14. This is troubling because the authors claim to have honed in on miR-83 based on the RNA-seq data.

-- We agree with your comments. Please note that in the RNA-seq results, *mir-83* is significantly increased in worms at day 21, and mildly upregulated at day 7 and day 14 although without statistical significance. This is why we claim to start studying *mir-83* from the RNA-Seq data and think the change of *mir-83* with age are generally similar in both RNA-Seq and qPCR results (i.e., upregulated in aged worms). Surely, we examined and confirmed this upregulation using multiple methods (including two transcriptional reporters) to draw this conclusion.

8. Figure 1. The western blots in Figure 1b are a bit concerning because the GFP blot is truncated and the α -tubulin blot appears saturated.

-- Thanks for your comments! We went over our western data and found that the α -tubulin blots were indeed saturated in one of the three replicates. Since we were also suggested to examine more time points, the corresponding experiments were re-

performed as shown in the revised Fig. 1b. The updated results still indicate an increase of *mir-83p::gfp* in aged worms. Please also see the uncropped blot in the Source Data.

9. Figure 1. The GFP images in Figure 1c suggest a much greater than 2-fold increase in GFP expression between day 7 and day 14.

-- Yes. The GFP signal from the intestine in Fig. 1c indeed shows a ~5-fold increase between day 7 and day 1. These quantification data are shown in Fig. 1d. Please note that this upregulation happens only in the intestine but not in neurons. That is why qPCR and western blot data from whole worms showed only a ~2-fold change.

10. Figure 1. The *hsf-1* RNAi treated animals in Figure 1f appear stunted in their development. The authors should note that development arrest or delay could indirectly affect miR-83 expression.

-- *hsf-1* RNAi is reported to reduce the worm size²⁰, as also seen in Fig. 1f, but does not cause development arrest or delay. By our observation, worms under *hsf-1* RNAi enter adulthood at ~60 hours post hatching, like those under control RNAi. Besides, the worms examined in Fig. 1f are of the same age (day 1 and day 7). So, we do not think that the reduced *mir-83* expression upon *hsf-1* RNAi is due to development arrest or delay.

11. Figure 4. It is not clear why over expressing miR-83 causes GFP-LGG-1 foci to increase to the same level as what is observed in the *mir-83* loss of function mutant in Figure 4e. Isn't this the opposite of what would be predicted? Same issue as in 4e.

-- We appreciate that this issue is noted. This is exactly the caveat of the GFP::*LGG-1*

reporter. The increase of its foci could mean either an increase (more autophagosome) or a decrease (no switch from autophagosome to autolysosome, and no degradation and recycling) of autophagy flux. That is why we used multiple methods to interrogate the autophagy. In the case of *mir-83(-)* mutants, both the number of GFP::LGG-1 puncta (APs) and mCherry::LGG-1 (ALs) were increased (Fig. 4a-b), indicating an elevated autophagy flux. In the case of *mir-83* overexpression, only GFP::LGG-1 puncta (APs) were increased whereas the AL number remains unchanged or even slightly reduced (Fig. 4e-f), indicating a blocked autophagy flux. A third autophagy reporter (SQST-1::GFP) further confirmed these phenotypes.

12. Figure S5. It's not clear why miR-83 is detected in the *mir-83* mutant line in Figure S5b.

-- Sorry for the confusion. We did not detect *mir-83* in the *mir-83(-)* mutants, but arbitrarily set the Ct value at 40 (qPCR reaction ends at cycle 40) to facilitate the quantification of relative transcription levels. The confusing data presentation has been corrected in the revised manuscript. Ct values are now presented instead of relative transcription levels.

13. Figure S5a cited out of order.

-- This issue has been addressed.

14. Without reading the methods section, it is unclear what is being measured in the miR-83 qPCR assays in Figures 6 and S5. In the results section and in the figures, when referring to the mature miR-83, the syntax miR-83 should be used. When referring to the *mir-83* gene, lower case italics should be used.

-- The manuscript has been modified accordingly. Thanks.

15. Lines 239-240. “...and the contamination from the two mir-83 enriched tissues (i.e., neurons and 240 intestine) was excluded.” It’s not clear what this means.

-- Sorry for the confusion. By examining the tissue-specific genes in isolated BWM cells, we did not detect genes specifically expressed in neurons (*rgef-1*) or in the intestine (*vha-6*), and therefore confirm that the isolated BWM cells were not contaminated with the neurons or intestinal cells. The corresponding description has been rephrased to further clarify this issue.

16. Methods: The specific antibodies, not just the manufacturers, should be noted.

-- The information has been included as suggested.

References

- 1 Shen, Y., Wollam, J., Magner, D., Karalay, O. & Antebi, A. A steroid receptor-microRNA switch regulates life span in response to signals from the gonad. *Science* **338**, 1472-1476, doi:10.1126/science.1228967 (2012).
- 2 Xu, Y., He, Z., Song, M., Zhou, Y. & Shen, Y. A microRNA switch controls dietary restriction-induced longevity through Wnt signaling. *EMBO reports*, doi:10.15252/embr.201846888 (2019).
- 3 Xiao, R. *et al.* RNAi Interrogation of Dietary Modulation of Development, Metabolism, Behavior, and Aging in *C. elegans*. *Cell reports* **11**, 1123-1133, doi:10.1016/j.celrep.2015.04.024 (2015).
- 4 Jan, C. H., Friedman, R. C., Ruby, J. G. & Bartel, D. P. Formation, regulation and evolution of *Caenorhabditis elegans* 3'UTRs. *Nature* **469**, 97-101, doi:10.1038/nature09616 (2011).
- 5 Hunt-Newbury, R. *et al.* High-throughput in vivo analysis of gene expression in *Caenorhabditis elegans*. *PLoS biology* **5**, e237, doi:10.1371/journal.pbio.0050237 (2007).
- 6 Martinez, N. J. *et al.* Genome-scale spatiotemporal analysis of *Caenorhabditis elegans* microRNA promoter activity. *Genome research* **18**, 2005-2015, doi:10.1101/gr.083055.108

- (2008).
- 7 Russell, J. C. *et al.* Isolation and characterization of extracellular vesicles from *Caenorhabditis elegans* for multi-omic analysis. *bioRxiv*, doi:10.1101/476226 (2018).
- 8 Chen, C. *et al.* Real-time quantification of microRNAs by stem-loop RT-PCR. *Nucleic acids research* **33**, e179, doi:10.1093/nar/gni178 (2005).
- 9 Wang, J. *et al.* *C. elegans* ciliated sensory neurons release extracellular vesicles that function in animal communication. *Current biology : CB* **24**, 519-525, doi:10.1016/j.cub.2014.01.002 (2014).
- 10 Melentijevic, I. *et al.* *C. elegans* neurons jettison protein aggregates and mitochondria under neurotoxic stress. *Nature* **542**, 367-371, doi:10.1038/nature21362 (2017).
- 11 Altun, Z. F. a. H., D.H. . Excretory system. *WormAtlas*, doi:10.3908/wormatlas.1.17 (2009).
- 12 Squadrito, M. L. *et al.* Endogenous RNAs modulate microRNA sorting to exosomes and transfer to acceptor cells. *Cell reports* **8**, 1432-1446, doi:10.1016/j.celrep.2014.07.035 (2014).
- 13 Kato, M., Chen, X., Inukai, S., Zhao, H. & Slack, F. J. Age-associated changes in expression of small, noncoding RNAs, including microRNAs, in *C. elegans*. *Rna* **17**, 1804-1820, doi:10.1261/rna.2714411 (2011).
- 14 Dzakah, E. E. *et al.* Loss of miR-83 extends lifespan and affects target gene expression in an age-dependent manner in *Caenorhabditis elegans*. *Journal of genetics and genomics = Yi chuan xue bao* **45**, 651-662, doi:10.1016/j.jgg.2018.11.003 (2018).
- 15 Aitlhadj, L. & Sturzenbaum, S. R. The use of FUdR can cause prolonged longevity in mutant nematodes. *Mechanisms of ageing and development* **131**, 364-365, doi:10.1016/j.mad.2010.03.002 (2010).
- 16 Feldman, N., Kosolapov, L. & Ben-Zvi, A. Fluorodeoxyuridine improves *Caenorhabditis elegans* proteostasis independent of reproduction onset. *PLoS one* **9**, e85964, doi:10.1371/journal.pone.0085964 (2014).
- 17 Rooney, J. P. *et al.* Effects of 5'-fluoro-2-deoxyuridine on mitochondrial biology in *Caenorhabditis elegans*. *Experimental gerontology* **56**, 69-76, doi:10.1016/j.exger.2014.03.021 (2014).
- 18 Burke, S. L., Hammell, M. & Ambros, V. Robust Distal Tip Cell Pathfinding in the Face of Temperature Stress Is Ensured by Two Conserved microRNAs in *Caenorhabditis elegans*. *Genetics* **200**, 1201-1218, doi:10.1534/genetics.115.179184 (2015).
- 19 Chang, J. T., Kumsta, C., Hellman, A. B., Adams, L. M. & Hansen, M. Spatiotemporal regulation of autophagy during *Caenorhabditis elegans* aging. *eLife* **6**, doi:10.7554/eLife.18459 (2017).
- 20 Walker, G. A., Thompson, F. J., Brawley, A., Scanlon, T. & Devaney, E. Heat shock factor functions at the convergence of the stress response and developmental pathways in *Caenorhabditis elegans*. *FASEB journal : official publication of the Federation of American Societies for Experimental Biology* **17**, 1960-1962, doi:10.1096/fj.03-0164fje (2003).

Reviewers' comments:

Reviewer #1 (Remarks to the Author):

In this revised manuscript, Zhou et al. have made great effort to address major concerns raised by all the reviewers, and have further improved their exciting study. The manuscript has also been extensively revised to elaborate on missing method/reagent details, novelty is more visible, and writing has been modified for better understanding. While in principle acceptable for publication, the authors should address the minor points listed below to improve the reading of the manuscript:

Scientific points:

1. The authors should indicate the full genotype for sqst-1 reporter in figure 3e (including promoter).
2. mir-83 levels represented in original Supp. Figure 5b are now referred to revised Figure 5b in review comments on Pg 8 & 9. Revised Figure number seems incorrect for inferred data.
3. It is unclear which representative lifespan is represented in Figure 7; this should be clearly indicated in Supp. Table 1 (which has been revised to show individual experiments).
4. Since the mechanism of microRNA transport was not probed, it would be appropriate to include a question mark on the depicted transport of microRNAs inside vesicles in Figure 7i.
5. Line 2: The authors are encouraged to include *C. elegans* in the title.
6. Line 366-372: The authors make claims about the Dzokhar paper's use of FUDR. The authors have no experimental evidence that this is the reason for the discrepancy. The authors are encouraged to simply acknowledge the differences in observation (including with citing other supporting papers) noting the experimental difference.
7. Line 271: Figure 1 does not formally show coelomocytes.
8. The authors need to include the methods for qPCR of sqst-1. How were animals aged and how many animals were used in this analysis?

Textual points:

- Line 24: "regulation on autophagy" should be "regulation of autophagy".
- Line 57-58: clarify what is meant with "a tissue-specific perturbation processes".
- Line 158-160: Fig. 3c does not measure autophagy flux.
- Line 235: D1 vs Day 1.
- Line 258: In should be from.
- Line 311-312 and 329: Emphasize in *C. elegans*.
- Line 361-363: While meaning is clear, syntax is incorrect.
- Correct to past tense: Lines 323, 326, 240, 260, and 303.
- Correct and delete 'the': Lines 79, 82, 84 (the to a), 165, 201, 207, 224, 282, and 364.
- Correct to insert 'the': Lines 344, 347.
- Correct for missing compound adjectives: Lines 30, 70, 80, 206, and 246.

Reviewer #2 (Remarks to the Author):

The authors have addressed my concerns and I do not mind my name being published alongside my review: Amy Buck

Reviewer #3 (Remarks to the Author):

How were the qPCR assays done such that they could distinguish between pri-miR-83 and pre-miR-83?

I don't think there's enough support for the conclusions that mature and not pre-miR-83 is transported across tissues. It certainly makes sense that this would be the case, but the evidence to support this is based on a qPCR assay for pre-miR-83 that is at the threshold of detection in whole animals and thus wouldn't be able to detect pre-miR-83 in BWMs given that even mature miR-83 levels are way lower in BWMs compared to whole animals (15 cycle difference). Even the pri-miR-83 transcript would not be predicted to be detectable in BWMs given its low abundance in whole animals.

I still can't find description of the RNA-seq experiments described in Figure S1.

Our detailed responses to the reviewers' concerns are listed below on a point-by-point basis.

Reviewer #1 (Remarks to the Author):

In this revised manuscript, Zhou et al. have made great effort to address major concerns raised by all the reviewers, and have further improved their exciting study. The manuscript has also been extensively revised to elaborate on missing method/reagent details, novelty is more visible, and writing has been modified for better understanding. While in principle acceptable for publication, the authors should address the minor points listed below to improve the reading of the manuscript:

Scientific points:

1. The authors should indicate the full genotype for sqst-1 reporter in figure 3e (including promoter).

-- Figure 3e has been modified according to your comment in the revised manuscript. Thanks for your suggestion!

2. mir-83 levels represented in original Supp. Figure 5b are now referred to revised Figure 5b in review comments on Pg 8 & 9. Revised Figure number seems incorrect for inferred data.

-- Sorry for the mistake in our previous rebuttal letter. As you have noticed, we were to refer to Fig. 6 about the miR-83 levels. No miR-83 was detected in *mir-83(-)* mutants.

3. It is unclear which representative lifespan is represented in Figure 7; this should be clearly indicated in Supp. Table 1 (which has been revised to show individual experiments).

-- Thank you for your suggestion. We now show in Supplementary Table 1 which replicate is presented as a representative lifespan assay.

4. Since the mechanism of microRNA transport was not probed, it would be appropriate to include a question mark on the depicted transport of microRNAs inside vesicles in Figure 7i.

-- The model figure (Figure 7i) has been modified as suggested.

5. Line 2: The authors are encouraged to include *C. elegans* in the title.

-- The title is modified as suggested.

6. Line 366-372: The authors make claims about the Dzokhar paper's use of FUDR. The authors have no experimental evidence that this is the reason for the discrepancy. The authors are encouraged to simply acknowledge the differences in observation (including with citing other supporting papers) noting the experimental difference.

-- We have already acknowledged the different observations in our lab (together with Slack lab) and by Dzokhar et al. in the "Discussion" section. But we also would like to make a speculation, albeit without experimental evidences, about the potential reason for this discrepancy. We believe that the readers can judge by themselves with our objective acknowledgement. If you think our speculation may obscure the readers' view, we are surely willing to remove it.

7. Line 271: Figure 1 does not formally show coelomocytes.

-- Thanks for your critics! The manuscript has been modified accordingly (Line 278).

8. The authors need to include the methods for qPCR of sqst-1. How were animals aged and how many animals were used in this analysis?

-- Thanks for your critical comments! We have clarified how animals aged in the “*C. elegans* strains and culture” and how many animals were used in other Method sections corresponding to specific assays.

Textual points:

- Line 24: “regulation on autophagy” should be “regulation of autophagy”.

- Line 57-58: clarify what is meant with “a tissue-specific perturbation process”.

- Line 158-160: Fig. 3c does not measure autophagy flux.

- Line 235: D1 vs Day 1.

- Line 258: In should be from.

- Line 311-312 and 329: Emphasize in *C. elegans*.

- Line 361-363: While meaning is clear, syntax is incorrect.

- Correct to past tense: Lines 323, 326, 240, 260, and 303.

- Correct and delete ‘the’ : Lines 79, 82, 84 (the to a), 165, 201, 207, 224, 282, and 364.

- Correct to insert 'the': Lines 344, 347.

- Correct for missing compound adjectives: Lines 30, 70, 80, 206, and 246.

-- All these issues have been corrected as suggested. Many thanks!

Reviewer #2 (Remarks to the Author):

The authors have addressed my concerns and I do not mind my name being published alongside my review: Amy Buck

-- Many thanks for your constructive comments and precious time reviewing our manuscript!

Reviewer #3 (Remarks to the Author):

How were the qPCR assays done such that they could distinguish between pri-miR-83 and pre-miR-83?

-- Many thanks for your critics!

We designed the qRT-PCR primers of pri-miR-83 and pre-miR-83 as previously described¹ (Fig. 1a for reviewer). The pri-RT primer is specific for pri-miR-83 target, whereas the pre-RT primer can reverse transcribe both pri- and pre-miR-83. As you kindly questioned, we did make a mistake in the previous version of the manuscript by taking the qRT-PCR signal from the pre-RT products as the amount of only pre-miR-83.

Figure 1 for reviewer. Zhou et al.

In theory, the amount of pre-miR-83 can be calculated by the difference between qRT-PCR signals of pre-RT products and pri-RT products (Fig. 1b for reviewer). The premise for this method is that the reverse transcription efficiencies of the pri-RT and pre-RT primers are exactly same. However, in the case of *mir-83*, we found this premise not working. The qRT-PCR signal of the pri-RT product is stronger but not weaker than that of the pre-RT product (Fig. 1b for reviewers). This is because the pre-RT primer binds to the hairpin of pre-miR-83 or pri-miR-83, which is less efficient than the binding of pri-RT primer to the linear part of pri-miR-83 (Fig. 1c for reviewers). The hairpin structure of the pre-RT products also affects the efficiency of the subsequent PCR reaction. Similar situations have been reported in mammal studies^{1,2}. qRT-PCR is therefore not a suitable method to quantify pri-miR-83 and pre-miR-83, as you pointed out.

To distinguish pre- and pri-miR-83, northern blot which is a regular method to examine absolute amounts of pri-, pre-, and mature microRNA should be performed^{3,4}. But it is technically too challenging for us to perform northern blot on single cells and EVs, due to the low amount of RNA from these samples. Because the focus of this manuscript is on whether *mir-83* is transported across tissues but not on its detailed species, we discarded the inaccurate qRT-PCR data of pre-miR-83, providing the readers with the information of pri- and mature miR-83 (Supplementary

Fig. 5 and 6).

Your critics have kept us from a serious mistake. Many thanks again!

I don't think there's enough support for the conclusions that mature and not pre-miR-83 is transported across tissues. It certainly makes sense that this would be the case, but the evidence to support this is based on a qPCR assay for pre-miR-83 that is at the threshold of detection in whole animals and thus wouldn't be able to detect pre-miR-83 in BWMs given that even mature miR-83 levels are way lower in BWMs compared to whole animals (15 cycle difference). Even the pri-miR-83 transcript would not be predicted to be detectable in BWMs given its low abundance in whole animals.

-- We fully agree with your comment. We ourselves do not think that only mature but not pri- or pre-miR-83 is transported across tissues. That is why we used "dominantly in its mature form" in the previous revised manuscript, which could be overstated. We think that due to the relatively low expression level of pri-miR-83 (as you noticed), much more mature miR-83 molecules must be transported across tissues than the pri-miR-83. Because of the inaccurate qRT-PCR results, we have no clear idea about the level of pre-miR-83 transported across tissues. The manuscript has been modified accordingly to clarify this issue. Please see the "Discussion" section.

I still can't find description of the RNA-seq experiments described in Figure S1.

-- Sorry for overlooking this issue. Corresponding description is included in the revised "Methods" section.

References

- 1 Schmittgen, T. D. *et al.* Real-time PCR quantification of precursor and mature microRNA. *Methods* **44**, 31-38, doi:10.1016/j.ymeth.2007.09.006 (2008).
- 2 Jiang, J., Lee, E. J., Gusev, Y. & Schmittgen, T. D. Real-time expression profiling of microRNA precursors in human cancer cell lines. *Nucleic acids research* **33**, 5394-5403, doi:10.1093/nar/gki863 (2005).
- 3 Liu, X. *et al.* A MicroRNA precursor surveillance system in quality control of MicroRNA synthesis. *Molecular cell* **55**, 868-879, doi:10.1016/j.molcel.2014.07.017 (2014).
- 4 Pawlicki, J. M. & Steitz, J. A. Primary microRNA transcript retention at sites of transcription leads to enhanced microRNA production. *The Journal of cell biology* **182**, 61-76, doi:10.1083/jcb.200803111 (2008).

REVIEWERS' COMMENTS:

Reviewer #1 (Remarks to the Author):

The authors have done a solid job addressing essentially all concerns, although this reviewer would reemphasize that the claims made in the Discussion about the Dzokhar paper should be toned down as the authors have no evidence to support the hypothesis that the findings in that paper is a side effect.

Moreover, the manuscript would benefit from being proofread by a native English speaker as many errors persist, e.g., first line of abstract, lines 79, 87, 100, 201, 204, 207, 213, 214, 220, 223, 253, 279, 280, 286, 328, 350, 357, 360, and 411. Moreover, authors should consistently use either Day 5 or D5.

Reviewer #3 (Remarks to the Author):

My concerns have been satisfactorily addressed in the revised manuscript.

Our detailed responses to the reviewers' comments are listed below on a point-by-point basis.

REVIEWERS' COMMENTS:

Reviewer #1 (Remarks to the Author):

The authors have done a solid job addressing essentially all concerns, although this reviewer would reemphasize that the claims made in the Discussion about the Dzokhar paper should be toned down as the authors have no evidence to support the hypothesis that the findings in that paper is a side effect.

-- We have removed our speculation as suggested.

Moreover, the manuscript would benefit from being proofread by a native English speaker as many errors persist, e.g., first line of abstract, lines 79, 87, 100, 201, 204, 207, 213, 214, 220, 223, 253, 279, 280, 286, 328, 350, 357, 360, and 411. Moreover, authors should consistently use either Day 5 or D5.

-- As suggested, we have sent our manuscript for English language editing services by NRES-NC. Besides, worm ages are now consistently labelled as "Day 5".

Reviewer #3 (Remarks to the Author):

My concerns have been satisfactorily addressed in the revised manuscript.

Many thanks to all reviewers for reviewing this manuscript and the constructive comments!